# Controllable and Constrained Sampling in Diffusion Models via Initial Noise Perturbation

Bowen Song[1]    Zecheng Zhang[2]    Zhaoxu Luo [1]    Jason Hu [1]    Wei Yuan[3]    Jing Jia[4]

Zhengxu Tang[1]          Guanyang Wang[3]*          Liyue Shen[1]*

[1]Department of Electrical Engineering and Computer Science, University of Michigan [2]TraceRoot.AI
[3]Department of Statistics, Rutgers University   [4]Department of Computer Science, Rutgers University

## Abstract

Diffusion models have emerged as powerful tools for generative tasks, producing high-quality outputs across diverse domains. However, how the generated data responds to the initial noise perturbation in diffusion models remains under-explored, which hinders understanding the controllability of the sampling process. In this work, we first observe an interesting phenomenon: the relationship between the change of generation outputs and the scale of initial noise perturbation is highly linear through the diffusion ODE sampling. Then we provide both theoretical and empirical study to justify this linearity property of this input-output (*noise-generation data*) relationship. Inspired by these new insights, we propose a novel *C*ontrollable and *C*onstrained *S*ampling (*CCS*) method, along with a new controller algorithm for diffusion models, that enables control over both the proximity of individual samples to a target image and the alignment of the sample mean with the target, while preserving good sample quality. We perform extensive experiments to compare our proposed sampling approach with other methods on both sampling controllability and sampled data quality. Results show that our CCS method achieves more precisely controlled sampling while maintaining superior sample quality and diversity, enhancing the applications of precise image editing. The code is available at *https://github.com/efzero/diffusioncontroller*.

## 1   Introduction

Recently, diffusion models achieve remarkable success in generative tasks such as text-to-image generation, audio synthesis [24, 31], as well as conditional generation tasks including inverse problem solving, image or video restoration, image editing, and translation [3, 28, 44, 6, 34, 9, 26, 37, 20, 45]. Despite these successes, real-world scientific and engineering problems pose more challenges on requesting reliable and controllable generation as well as data privacy.

To tackle this, one important question is: *How to control the distribution of samples from a diffusion model to match a specific target?* Previous works on controllable generation with diffusion models mostly focus on constraining the generation process sample-by-sample using either plug-and-play approaches [29, 9, 26, 37] or modifying the unconditional score [31, 45, 16, 6], so that each sample can satisfy a measurement constraint. However, most prior works focus on per-sample control, with limited exploration of how to regulate the overall distribution of generated samples to meet specific

---

*Joint senior authors.
Correspondence: Bowen Song (bowenbw@umich.edu), Guanyang Wang (guanyang.wang@rutgers.edu), and Liyue Shen (liyues@umich.edu)

statistical constraints, which is a crucial requirement in differential privacy [15]. This inspires the novel task for controllable and constrained sampling we are targeting in this paper. Considering the unique mechanism in diffusion sampling, we are motivated to exploit the initial noise control by studying this key question: *How do the initial noise perturbations affect the generated samples in diffusion models?* Previous works [2, 41] suggest that the learned posterior mean predictor function is locally linear with perturbation among a certain range of timesteps for diffusion models. However, this linearity cannot be applied to every timestep nor to the samples of diffusion models. From a new perspective, this work sheds lights on the relationship between input noise perturbations and generation data in diffusion models, by proposing a training-free approach.

First of all, we observe an interesting phenomenon that when using denoising diffusion implicit models (DDIM) sampling, the initial noise has a highly linear effect on the generation data at small or moderate scales. Motivated by this observation, our study tries to justify this linearity property via initial noise perturbation theoretically and empirically.

Based on the spherical interpolation to perturb the initial noise vector, we propose a novel **C**ontrollable and **C**onstrained **S**ampling method (**CCS**) for diffusion models to sample with a target rMSE level, enabling the sample mean to be close to the target image, while preserving high quality and adjustable diversity. The motivation for this task stems from a fundamental need in image editing and controllable generation: preserving key source features while allowing controlled variation. However, few studies benchmark sample quality and key feature preservation at a target controlled variation level. Our first key idea is to fix the average distance (rMSE) between samples and the target, enabling a fair comparison of sample diversity and feature preservation. Our second insight is to evaluate the distance between the sample mean and the target image, which reveals how well common features are preserved. In addition, our CCS algorithm enables a user-controllable "diversity slider": a tool that adjusts how far generated samples deviate from the input image. This fine-grained control over similarity can be vital for practical applications such as photo editing apps.

Furthermore, we conduct extensive experiments to validate the linearity phenomenon and then investigate the controllability performance of our proposed CCS method by generating images centered around a specified target mean image with a certain distance. Results demonstrate the superiority of our CCS method in both controllability and sampled image quality compared with baseline methods. Moreover, we show the potential of proposed CCS sampling for broader applications including precise image editing.

Our contributions can be summarized as below:

- We unveil a novel linear relationship between the initial noise and generated samples for DDIM sampling. We justify it theoretically, validate it thoroughly through extensive experiments, and discuss practical implications.

- We propose a novel task of controllable generation with the goal of making sample mean close to a target mean while controlling the MSE of samples to a target level. To the best of our knowledge, we are the first to study this task. This task can be useful for benchmarking the performance of personalized image generation.

- We propose a novel controllable sampling method based on our discovered linearity relationship. Extensive experiments with both pixel and latent diffusion models demonstrate the superior performance of our algorithm in achieving precise controllability within a our proposed constrained sampling framework.

## 2  Background

**Diffusion Models.**  Diffusion models consists of a forward process that gradually adds noise to a clean image, and a reverse process that denoises the noisy images [35, 38]. The forward model is given by $\mathbf{x}_t = \mathbf{x}_{t-1} - 0.5\beta_t\Delta t\mathbf{x}_{t-1} + \sqrt{\beta_t}\Delta t\omega$ where $\omega \in \mathbb{N}(0, I)$ and $\beta(t)$ is the noise schedule of the process. The distribution of $\mathbf{x}_0$ is the clean data distribution, while the distribution of $\mathbf{x}_T$ is approximately a standard Gaussian distribution. When we set $\Delta t \to 0$, the forward model becomes $\mathrm{d}\mathbf{x}_t = -0.5\beta_t\mathbf{x}_t\mathrm{d}t + \sqrt{\beta_t}\mathrm{d}\omega_t$, which is a stochastic differential equation (SDE). The reverse of this SDE is given by:

$$\mathrm{d}x_t = \left(-\frac{\beta(t)}{2} - \beta(t)\nabla_{\mathbf{x}_t}\log p_t(\mathbf{x}_t)\right)dt + \sqrt{\beta(t)}d\overline{\omega}.$$

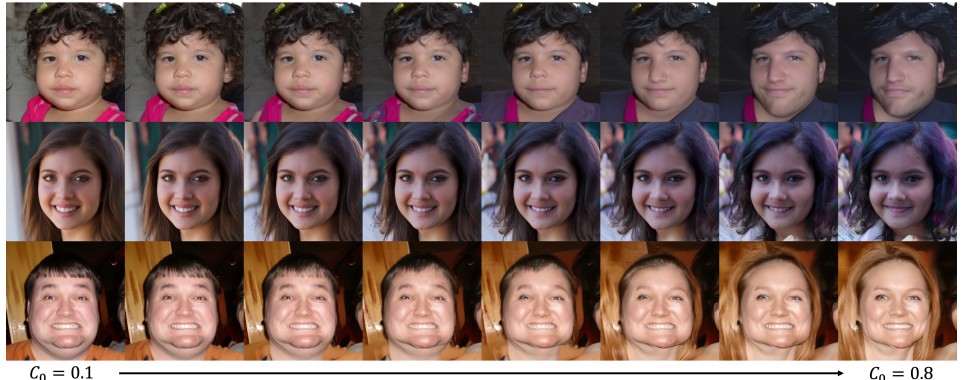

$C_0 = 0.1$ ⟶ $C_0 = 0.8$

Figure 1: Qualitative demonstration of linearity when increasing scale of perturbation. For each target mean, we sample a perturbation noise and gradually increase $C_0$ (0.1 at a time) to increase the magnitude of the perturbation.

One can train a neural network to learn the score function $\nabla_{\mathbf{x}_t} \log p_t(\mathbf{x}_t)$. However, this formulation involves running many timesteps with high randomness. We can also compute the equivalent Ordinary Differential Equation (ODE) form to the SDE, which has the same marginal distribution of $p(\mathbf{x}_t)$. A sampling process, called denoising diffusion implicit models (DDIM), modifies the forward process to be non-markovian, so as to form a deterministic probability-flow ODE for the reverse process [36]. In this way, we are able to achieve significant speed-up sampling. More discussion on this can be found in Section 3.

**Constrained Generation with Diffusion Models.** Constrained generation requires to sample $\mathbf{x}_0$ subject to certain conditions or measurements $\mathbf{y}$. The conditional score at $T$ can be computed by the Bayes rule, such that

$$\nabla_{\mathbf{x}_t} \log p_t(\mathbf{x}_t | \mathbf{y}) = \nabla_{\mathbf{x}_t} \log p_t(\mathbf{x}_t) + \nabla_{\mathbf{x}_t} \log p_t(\mathbf{y} | \mathbf{x}_t). \tag{1}$$

The second term can be computed through classifier guidance [13], where an external classifier is trained for $p_0(\mathbf{y}|\mathbf{x}_0)$ or $p_t(\mathbf{y}|\mathbf{x}_t)$, and then can be plugged into the diffusion model through Eq. 1. Diffusion posterior sampling [6] further refines this formulation by proposing to perform posterior sampling with the approximation of $p(\mathbf{y}|\mathbf{x}_t) \approx p(\mathbf{y}|\hat{\mathbf{x}}_0)$, where $\hat{\mathbf{x}}_0$ is the Minimum Mean Square Error (MMSE) estimator of $\mathbf{x}_0$ based on $\mathbf{x}_t$.

Another line of works exploit hard consistency, which projects the intermediate noise to a measurement-consistent space during sampling via optimization and plug-and-play [7, 9, 29, 34]. However, the projection term can damage the sample quality [6]. However, these works all target on controlling each individual sample. To the best of our knowledge, few works explore how to control the distribution of generated samples to match certain statistical constraints, such as centered around a specified target mean with certain distance, which is the target for this work.

**Noise Perturbation in Diffusion Models.** Noise adjustment for diffusion models has been explored in image editing, video generation, and other applications [28, 44, 10, 18, 42, 46] for changing the style or other properties of the generated data. However, a principled study on how the noise adjustment affects the samples is limited in diffusion models. Recently, [2, 41] observe the local linearity and low-rankness of the posterior mean predictor $\hat{\mathbf{x}}_0$ based on $\mathbf{x}_t$ in large timesteps, but this study cannot extend to the analysis of generated samples. In this work, we investigate how initial noise perturbations affect generated samples from the diffusion model in the ODE sampling setting.

## 3 Linear Relationship between Initial Noise and Outputs in Diffusion Models

This section analyzes how small perturbations in the input noise affect the generation data under the DDIM sampling framework. We show that a slight change in the initial noise leads to an approximately linear variation in the sampled images. This result is quantified from two perspectives: the discretized DDIM sampling process [36] and the associated continuous-time ODE. Our mathematical analysis

relies on minimal assumptions, which also serves as the foundation for our proposed CCS algorithm in Section 4.

## 3.1 Preliminary: DDIM Sampling

Fix the total sampling timesteps $T$ and an initialization noise sample $\mathbf{x}_T$, [36] generates samples from the backward process $\mathbf{x}_T \to \mathbf{x}_{T-1} \to \ldots \to \mathbf{x}_0$ using the following recursive formula:

$$\mathbf{x}_{t-1} = \sqrt{\alpha_{t-1}} \left( \frac{\mathbf{x}_t - \sqrt{1 - \alpha_t}\, \epsilon_\theta^{(t)}(\mathbf{x}_t)}{\sqrt{\alpha_t}} \right) + \sqrt{1 - \alpha_{t-1} - \sigma_t^2}\, \epsilon_\theta^{(t)}(\mathbf{x}_t) + \sigma_t \epsilon_t, \qquad (2)$$

where $\alpha_t$ corresponds to the noise schedule in DDPM, $\epsilon_\theta^{(t)}(\mathbf{x}_t)$ is the predicted noise given by the pre-trained neural network with parameter $\theta$, $\epsilon_t$ is the standard Gaussian noise, and $\sigma_t$ is a hyperparameter. The DDIM sampler [36] sets $\sigma_t = 0$ to make the backward process deterministic once $\mathbf{x}_T$ is fixed. It is known (*e.g.*, eq (11) of [14]) that predicting the noise is equivalent to predicting the score function up to a normalizing factor, *i.e.*, $\epsilon_\theta^{(t)}(\mathbf{x}_t) \approx -\sqrt{1 - \alpha_t} \nabla_{\mathbf{x}_t} \log p_t(\mathbf{x}_t)$. By setting $\sigma_t = 0$ and substituting $\epsilon_\theta^{(t)}$ with its corresponding estimand, we obtain the *idealized DDIM process*:

$$\mathbf{x}_{t-1} = \sqrt{\alpha_{t-1}} \left( \frac{\mathbf{x}_t + (1 - \alpha_t) \nabla_{\mathbf{x}_t} \log p_t(\mathbf{x}_t)}{\sqrt{\alpha_t}} \right) - \sqrt{(1 - \alpha_{t-1})(1 - \alpha_t)} \nabla_{\mathbf{x}_t} \log p_t(\mathbf{x}_t). \qquad (3)$$

If we treat the index $t$ as a continuous variable (and rewrite $\alpha_t$ as $\alpha(t)$ to avoid confusion), we can write the idealized ODE as:

$$d\bar{\mathbf{x}}_t = -\sqrt{1 - \alpha(t)} \nabla \log p_t \left( \frac{\bar{\mathbf{x}}_t}{\sqrt{\sigma^2(t) + 1}} \right) d\sigma(t). \qquad (4)$$

We now examine how a small perturbation $\mathbf{x}_T \to \mathbf{x}_T + \lambda \Delta \mathbf{x}$ would affect the output sample at time $t = 0$ through both the discrete (3) and continuous time (4) perspectives.

**Related work:** Theorem 1 in [3] presents a related result on the impact of initial noise perturbation. Our study differs from theirs in a variety of aspects. Firstly, they study $\mathbb{E}[\mathbf{x}_0 \mid \mathbf{x}_t + \lambda \Delta \mathbf{x}] - \mathbb{E}[\mathbf{x}_0 \mid \mathbf{x}_t]$ under the (stochastic) diffusion process. In contrast, we directly examine the output $\mathbf{x}_0$ given the initializations $\mathbf{x}_t$ and $\mathbf{x}_t + \lambda \Delta \mathbf{x}$ under the deterministic DDIM (3) or the ODE process (4). Secondly, [3] assumes that $p_0$ is a low-rank mixture of Gaussian distributions, which allows for an analytical solution for $p_t$. In contrast, our weaker assumptions render $p_t$ analytically intractable. Consequently, we use very different techniques, such as ODE stability theory and Grönwall's inequality, to study the system's behavior.

## 3.2 Linearity in DDIM Discretized Sampling

Previous works reveal the local linearity of the denoiser (which learns the score function) is quite strong [2, 27] in certain range of timesteps for diffusion models. Indeed, we can demonstrate that at very large noise levels, the score function is approximately linear. If a distribution is Gaussian, its score function is a linear function. Let: $\mathbf{x} \sim \mathcal{N}(\mu, \Sigma)$ the score function is given by: $\nabla_{\mathbf{x}} \log p(\mathbf{x}) = -\Sigma^{-1}(\mathbf{x} - \mu)$ which is linear in $\mathbf{x}$.

This explains why the denoiser exhibits high linearity at large timesteps as observed in [2, 27]. For DDPM, since $\mathbf{x}_t = \sqrt{\alpha_t} \mathbf{x}_0 + \sqrt{1 - \alpha_t} \epsilon$ where $\epsilon \sim \mathcal{N}(0, I)$, as $t$ increases, the noise term dominates, and $p(\mathbf{x}_t)$ approaches a Gaussian. Based on this observation, we can derive an approximately linear relationship between change in the input initial noise and output of DDIM sampling as demonstrated in Proposition 1.

**Proposition 1.** *With all the notations defined as above, assuming* $\log p_t$ *is second-order differentiable for every* $t \geq 1$, *there exists a matrix-valued function* $\gamma_0$ *such that*

$$\mathbf{x}_0(\mathbf{x}_T + \lambda \Delta \mathbf{x}, T) = \mathbf{x}_0(\mathbf{x}_T, T) + \lambda \gamma_0(\mathbf{x}_T) \Delta \mathbf{x} + o(\lambda).$$

*In turn,*

$$\|\mathbf{x}_0(\mathbf{x}_T + \lambda \Delta \mathbf{x}, T) - \mathbf{x}_0(\mathbf{x}_T, T)\|_2 = \|\lambda \gamma_0(\mathbf{x}_T) \Delta \mathbf{x}\|_2 + o(\lambda).$$

Proposition 1 shows that a linear perturbation of the input with magnitude $\lambda$ and direction $\Delta\mathbf{x}$ results in an approximately linear change in the output, with magnitude $|\lambda|\|\gamma_0(\mathbf{x}_T)\Delta\mathbf{x}\|_2$ and direction $\gamma_0(\mathbf{x}_T)\Delta\mathbf{x}$. Recalling Eq. 2, each idealized DDIM sampling can be viewed as a linear combination of the current intermediate noisy input $\mathbf{x}_t$ and the score function $\nabla_{\mathbf{x}_t}\log p_t(\mathbf{x})$. Based on this observation, Proposition 1 can be derived recursively using the linear approximation of the score function since each DDIM sampling step takes a linear combination of the predicted score and the intermediate noise. The derivation can be found in Appendix A.1. Our assumption is based solely on the second-order smoothness of the score, which is weaker than most existing assumptions depending on the data distribution $p_0$. For example, our assumptions hold under common conditions in the literature, such as the manifold hypothesis [11, 38] or the mixture of (low-rank) Gaussian assumption [17, 3, 1].

Furthermore, at large $t$, $p_t$ is approximately Gaussian and $\nabla_{\mathbf{x}_t}\log p_t(\mathbf{x}_t)$ is smooth, which implies small linear approximation error. The reason for the small error is that: 1) the score function of a Gaussian is linear in $\mathbf{x}$ by $\nabla_{\mathbf{x}}\log p(\mathbf{x}) = -\Sigma^{-1}(\mathbf{x} - \mu)$, 2) when a function is smooth and its higher-order derivatives are small in magnitude, it has fewer abrupt changes, and the linear approximation error is bounded by the norm of the Hessian of the score function (through the Taylor Remainder Theorem), leading to low linear approximation error. However, one might be concerned that the linear approximation error could grow significantly when $t$ decreases and $p_0$ contains multiple clusters with low-density regions in between. Nevertheless, we now explain why this concern does not arise in practice. The coefficient $f(t) := -\sqrt{\alpha_t}^{-1}\sqrt{\alpha_{t-1}(1-\alpha_t)} + \sqrt{1-\alpha_{t-1}}$ of $\epsilon_\theta^{(t)}(\mathbf{x}_t)$ in (2) is close to 0 for small $t$, as $\alpha_t \approx 1$. Moreover, the structure of the neural network $\epsilon_\theta$ ensures that the output is normalized and bounded in norm, so the change in output is also bounded. Consequently, for a small perturbation in $\mathbf{x}_t$, we have $\|f(t)(\epsilon_\theta^{(t)}(\mathbf{x}_t + \Delta\mathbf{x}) - \epsilon_\theta^{(t)}(\mathbf{x}_t))\|_2 \approx 0$ when $t$ is small.

**Linear Approximation Error.** We provide further analysis of this linear approximation error in the Appendix. We derive that this error is affected by the magnitude and the smoothness of local probability density $p(x_0 = x_{\text{sample}})$.

### 3.3 ODE Stability

Let $\bar{\mathbf{x}}_0(\mathbf{x}, T)$ be the solution of (4) with initialization $\mathbf{x}_T = \mathbf{x}$ (i.e., $\bar{\mathbf{x}}_T = \mathbf{x}/\sqrt{\alpha(T)}$) at timestep $T$, and $x_0(\mathbf{x}, T) = \alpha(0)\bar{\mathbf{x}}_0(\mathbf{x}, T)$. With some technical assumptions that is detailed in Appendix, we have the following:

**Proposition 2.** *There exists a matrix-valued function $\psi_0$ such that:*
$$\bar{\mathbf{x}}_0(\mathbf{x}_T + \lambda\Delta\mathbf{x}, T) = \bar{\mathbf{x}}_0(\mathbf{x}_T, T) + \lambda\psi_0(\mathbf{x}_T)\Delta\mathbf{x} + o(\lambda).$$

*In turn,*
$$\mathbf{x}_0(\mathbf{x}_T + \lambda\Delta\mathbf{x}, T) = \mathbf{x}_0(\mathbf{x}_T, T) + \lambda\sqrt{\alpha(0)}\psi_0(\mathbf{x}_T)\Delta\mathbf{x} + o(\lambda).$$

Proposition 2 mirrors Proposition 1 but is formulated in the continuous-time ODE setting. Its proof relies on ODE stability theory, showing that the output change is "approximately linear" for sufficiently small $\lambda$. Furthermore, under the same assumption, we establish that the change remains "at most linear" for all $\lambda$. The proof, which applies Grönwall's inequality, is provided in Appendix.

**Proposition 3.** *With the same assumptions as above, there exists a constant $C(T)$ depending on $T$ such that for any $\lambda$:*
$$\|\bar{\mathbf{x}}_0(\mathbf{x}_T + \lambda\Delta\mathbf{x}, T) - \bar{\mathbf{x}}_0(\mathbf{x}_T, T)\|_2 \leq C(T)|\lambda|\|\Delta\mathbf{x}\|_2.$$

## 4 Sampling with Control

Our objective is to perturb $\mathbf{x}_T$ into a *random* $\mathbf{x}_T'$ such that the generated image $\mathbf{x}_0'$ such that it has *1. a sample mean close to $\mathbf{x}_0$ while maintaining 2. sufficient diversity and difference from the original image* and *3. high image quality*. We preserve the notation $\mathbf{x}_0$ to denote a "target image" or "target mean". We also preserve the notation $\mathbf{x}_T := \text{DDIM}^{-1}(\mathbf{x}_0; 0, T)$, the "noise" by finding a reliable initial noise $\mathbf{x}_T$, such that $\text{DDIM}(\mathbf{x}_T) = \mathbf{x}_0$ The closeness is quantified by $L$-2 norm distance $\|\mathbb{E}[\mathbf{x}_0'] - \mathbf{x}_0\|_2$, and the diversity is measured by $\mathbb{E}[\|\mathbf{x}_0' - \mathbf{x}_0\|_2^2]$. A notable feature of our algorithm is that users can specify a desired level of diversity (such as using $C_0$ in Fig.1), and the generated images will match this level while ensuring $\mathbb{E}[\mathbf{x}_0'] \approx \mathbf{x}_0$. Our mechanism is defined as $\mathbf{x}_T' = a\mathbf{x}_T + b\Delta$, where $\Delta$ is a random perturbation, and $a$ and $b$ are parameters to be specified shortly.

## 4.1 Sampling around a Center

For an input of the form $\mathbf{x}'_T = a\mathbf{x}_T + b\Delta$ with random $\Delta$, when $b$ is small and $a$ is close to 1, it can be regarded as a slight perturbation of $\mathbf{x}_T$. Based on Section 3, the output will remain close to $\mathbf{x}_0$ with an additional linear adjustment applied to $b\mathbf{n}$. Thus, we define $\hat{\mathbf{x}}'_0 := \mathbf{x}_0 + bA\Delta$ as an approximation for $\mathbf{x}'_0$, where $A = \gamma_0(a\mathbf{x}_T + b\Delta)$ specified in Proposition 1. Since $\Delta$ is the only source of randomness in $\hat{\mathbf{x}}'_0$, we can easily calculate $\mathbb{E}[\hat{\mathbf{x}}'_0] = \mathbf{x}_0 + bAE[\Delta]$ and $\mathrm{Var}[\hat{\mathbf{x}}'_0] = b^2 A\,\mathrm{Cov}(\Delta)A^\top$. We will now discuss the principles for our sampling design.

**High-quality image generation:** we first note that the input to both DDPM and DDIM samplers is standard Gaussian noise. The following feature is known as the "concentration phenomenon" of a high-dimensional Gaussian:

**Proposition 4.** *Let $X \sim \mathbb{N}(0, I_d)$, then for any $\delta \in (0, 1)$*

$$\mathbb{P}\left[\|X\|_2^2 \in (1 \pm \delta)d|\right] \geq 1 - 2\exp\left(-\frac{1}{2}d\left(\frac{1}{2}\delta^2 - \frac{1}{3}\delta^3\right)\right).$$

This result suggests that a standard Gaussian noise vector remains close to a hypersphere of radius $\sqrt{d}$.

**Close to target mean:** Our approximation $\hat{\mathbf{x}}'_0$ has expectation $\mathbb{E}[\hat{\mathbf{x}}'_0] = \mathbf{x}_0 + bAE[\Delta]$. Thus, it is sufficient to select $\Delta$ such that $\mathbb{E}[\Delta] = 0$ in order to achieve: $\mathbb{E}[\hat{\mathbf{x}}_0] \approx \mathbb{E}[\hat{\mathbf{x}}'_0] = \mathbf{x}_0$, where the first approximation is justified by Proposition 1 and 2, with further empirical validation in Appendix.

## 4.2 Centering Feasibility

The simplest strategy is to add a random noise vector $\Delta\mathbf{x}$ directly to $\mathbf{x}_T$, expressed as $\mathbf{x}'_T = \mathbf{x}_T + \Delta\mathbf{x}$ (with $a = 1, b\Delta = \Delta\mathbf{x}$). However, the following proposition demonstrates that this approach cannot produce high-quality images.

**Proposition 5.** *For any fixed vector $\mathbf{x}$, and any random vector $\Delta\mathbf{x}$ such that $\mathbb{E}[\Delta\mathbf{x}] = 0$, the following holds:*
$$\mathbb{E}[\|\mathbf{x} + \Delta\mathbf{x}\|_2^2] = \|\mathbf{x}\|_2^2 + \mathrm{tr}(\mathrm{Cov}[\Delta\mathbf{x}]) \geq \|\mathbf{x}\|_2^2,$$
*with equality if and only if $\Delta\mathbf{x} = 0$ almost surely.*

Proposition 5 indicates that directly adding noise, $\mathbf{x}_T \to \mathbf{x}'_T := \mathbf{x}_T + \Delta\mathbf{x}$, pushes $\mathbf{x}'_T$ farther from the spherical surface. This partly explains why the average image becomes blurrier or noisier as the scale of $\Delta\mathbf{x}$ increases, since the drift term $\mathrm{tr}(\mathrm{Cov}[\Delta\mathbf{x}])$ grows larger, causing $\mathbf{x}'_T$ to deviate further from the sphere with radius $\|\mathbf{x}_T\|_2$. On the other hand, a simple linear interpolation such as for also cannot produce high-quality images", because this will shrink the magnitude of the interpolated vector, which we demonstrate in the experiments. This inspires us to consider the spherical linear interpolation method [33] for sampling, as described below. Similar approaches have been proposed by [46, 35], but only for interpolating between two images.

## 4.3 Spherical Interpolation

Let vectors $\mathbf{a}$ and $\mathbf{b}$ satisfy $\|\mathbf{a}\|_2 = \|\mathbf{b}\|_2$ and form an angle $\theta$. Then for any $\alpha \in (0, 1)$, the vector obtained through spherical interpolation $\mathbf{c} := \frac{\sin(\alpha\theta)}{\sin\theta}\mathbf{a} + \frac{\sin((1-\alpha)\theta)}{\sin(\theta)}\mathbf{b}$ satisfies $\|\mathbf{c}\|_2 = \|\mathbf{a}\|_2 = \|\mathbf{b}\|_2$. In our case, for a standard $d$-dimensional normal noise vector $\epsilon$, it is known $\|\epsilon\|_2 \approx \sqrt{d} \approx \|\mathbf{x}_T\|_2$. Therefore, we can do spherical interpolation between $\mathbf{x}_T$ and $\epsilon$ to obtain $\mathbf{x}'_T$. Our CCS algorithm is described in Algorithm 1.

The perturbation mechanism corresponds to $\mathbf{x}'_T = a\mathbf{x}_T + b\Delta$ with $a = \sin(\theta - C_0)/\sin(\theta), b = \sin(C_0)/\sin(\theta)$, and $\Delta$ is a

---

**Algorithm 2** Controller Tuning (CT)

1: **Input:** target mean $\mathbf{x}_0$, target diversity level $\mathrm{MSE}_{\text{target}}$, tolerance: tol, $C_0$, and $C_{high}$
2: **Initialize:** $C_0 \leftarrow \frac{C_{\text{low}} + C_{\text{high}}}{2}$
2: **while** not converged **do**
3:     Sample a batch of $\mathbf{x}'_0$ by Alg. 1
4:     **if** $|\mathbb{E}[\|\mathbf{x}'_0 - \mathbf{x}_0\|_2] - \mathrm{MSE}_{\text{target}}| < \text{tol}$ **then**
5:       **Break**
6:     **else if** $\mathbb{E}[\|\mathbf{x}'_0 - \mathbf{x}_0\|_2] > \mathrm{MSE}_{\text{target}}$ **then**
7:       $(C_{\text{high}}, C_0) \leftarrow (C_0, \frac{C_0 + C_{\text{low}}}{2})$
8:     **else**
9:       $(C_{\text{low}}, C_0) \leftarrow (C_0, \frac{C_0 + C_{\text{high}}}{2})$
10:     **end if**
10: **end while**=0

---

**Algorithm 1** (Full Inversion) CCS Sampling

---

**Requires:** target mean $\mathbf{x}_0$, perturbation scale $C_0$, number of diffusion model timesteps $T$

**Step 0:** Compute the DDIM inversion of $\mathbf{x}_0$, i.e. $\mathbf{x}_T = \text{DDIM}^{-1}(\mathbf{x}_0, 0, T)$

**Step 1:** Sample noise $\epsilon \sim \mathbb{N}(0, I)$. Then compute

$$\theta = \cos^{-1}\left(\frac{\epsilon \cdot \mathbf{x}_T}{||\epsilon||_2 ||\mathbf{x}_T||_2}\right)$$

**Step 2:** Compute $\mathbf{x}'_T$ using spherical interpolation formula:

$$\mathbf{x}'_T = \frac{\sin(C_0)}{\sin(\theta)} \cdot \epsilon + \frac{\sin(\theta - C_0)}{\sin(\theta)} \cdot \mathbf{x}_T$$

**Step 3:** Output sample $\mathbf{x}'_0 = \text{DDIM}(\mathbf{x}'_T, T, 0)$

---

standard Gaussian noise. $C_0 := \alpha\theta$ is defined as the parameter of perturbation scale. This mechanism satisfies the design principles described in Section 4.1: $\mathbb{E}[\epsilon] = 0$ ensures that the new sample remains close to the target mean, while the Gaussian concentration and spherical interpolation ensure that $||\mathbf{x}'_T||_2 \approx ||\mathbf{x}_T||_2$, resulting in high-quality generated images. Parameter $C_0$ controls sampling diversity. In the extreme case $C_0 = 0$, we have $\mathbf{x}'_T = \mathbf{x}_T$, so $\mathbf{x}'_0$ matches $\mathbf{x}_0$ exactly but has no diversity. A larger $C_0$ makes the perturbed input deviate more from the original image and gets closer to noise. This leads to greater diversity in the generated image.

Algorithm 2 allows users to control the desired level of diversity. It works by calling Alg. 1 for different values of $C_0$, which are determined through binary search. The process is repeated until the desired diversity level (up to a small tolerance threshold) is reached: if the MSE of generated images to target mean is below target threshold, $C_0$ is increased; otherwise, it is decreased.

The following theorem demonstrates that the CCS algorithm is able to precisely control the input distance.

**Proposition 6.** *Denote the dimensionality of $\mathbf{x}_T$ by $d$. Given an initial noise $\mathbf{x}_T$ with $||\mathbf{x}_T||_2 = (1 + o(1))\sqrt{d}$, and fix a small $\delta > 0$. For any $M \leq (2 - \delta)\sqrt{d}$, then we can find $C_0$ in Algorithm 1 such that with probability $p_d \to 1$ as $d \to \infty$, we have $||\mathbf{x}'_T - \mathbf{x}_T||_2 = M$.*

Since the dimensionality of our problem is sufficiently large, Proposition 6 allows users to control $M$ as the input distance. Consequently, Algorithm 1 can generate a random interpolants with an exact distance of $M$ from the input. Furthermore, since the direction is uniformly distributed, and when $C_0$ is small, $\mathbb{E}[\mathbf{x}'_T] \approx \mathbb{E}[\mathbf{x}_T]$, and $\mathbb{E}[\mathbf{x}'_0] \approx \mathbb{E}[\mathbf{x}_0]$, which satisfies our design goal.

In other cases when the inverted noise does not lie on the standard Gaussian hypersphere, we argue that our proposed spherical interpolation leads the second moment closer to a standard Gaussian. Formally, Fix any vector $\mathbf{x} \in \mathbb{R}^d$, and let $\epsilon \sim \mathbb{N}(0, I_d)$. Let $\theta \in [0, \pi]$ be the angle between $\mathbf{x}$ and $\epsilon$. We define the interpolated vector:

$$\mathbf{y} = \frac{\sin(c\theta)}{\sin(\theta)}\epsilon + \frac{\sin((1-c)\theta)}{\sin\theta}\mathbf{x}.$$

Our goal is to show $\mathbf{y}$ is closer than $\mathbf{x}$ to a Gaussian in the second-moment (energy-shell) sense. Since a standard Gaussian has second moment $\mathbb{E}[||Z||^2] = d$, we define the gap of second moment as:

$$\delta(Y) := |\mathbb{E}[Y^2] - d|.$$

**Proposition 7.** *For any $c \in (0, 1)$, we have:*

$$\delta(\mathbf{y}) \leq \delta(\mathbf{x}).$$

In summary, we argued that we can center our samples around the target mean better through spherical interpolation with random noise as in Prop. 5, and control the distance to the sample mean through adjusting the perturbation scale $C_0$ as in Prop. 6, and we can also improves the sample quality even if the initial noise is not on a Gaussian hypersphere as demonstrated in Prop. 7.

## 4.4 Extension to Conditional Latent Diffusion Models

Conditional diffusion models usually compute the conditional score with classifier-free guidance (CFG). Let $s_\theta(\mathbf{x}_t, t)$ be the predicted noise, it can be written in $s_\theta(\mathbf{x}_t, t) = s_\theta(\mathbf{x}_t, t, c_{null}) + \gamma(s_\theta(\mathbf{x}_t, t, c) - s_\theta(\mathbf{x}_t, t, c_{null}))$ where $\gamma$ is the CFG term, $c$ is the condition and $c_{null}$ is the null condition. The computation is more expensive, and we may not want to change the semantics drastically by a small perturbation. Motivated by this, we propose a Partial-Inversion CCS Sampling algorithm (P-CCS). Instead of starting from the $T$, we pick an intermediate timestep $t_0$. Then, we compute the noise term from DDIM inversion by subtracting the clean component, sample a new noise from $\mathbb{N}(0, (1 - \alpha_{t_0})I)$, and then perform spherical interpolation. Details of this partial inversion algorithm (P-CCS) can be found in the Appendix.

## 5 Applications and Experiments

In this section, we will discuss the applications of the observed linearity property with the proposed controllable sampling techniques.

### 5.1 Linearity Property

**Experimental Validation.** We perform extensive experiments on both pixel diffusion models on the FFHQ [22] and CIFAR-10 [25] dataset and latent diffusion models on the Celeba-HQ and fMoW dataset [5].

Table 1: $R^2$ between input perturbation and normalized residual norms

| Pixel Diffusion Models | | Latent Diffusion Models | |
|---|---|---|---|
| FFHQ | CIFAR-10 | CelebA-HQ | fMoW |
| 0.995 | 0.988 | 0.959 | 0.947 |

For each experiment, we first sample 50 images as target images from each validation dataset from FFHQ, CIFAR-10, and Celeba-HQ. We also pick one images from each class from the validation set of the fMoW dataset for further verification. Then for the FFHQ and CIFAR-10 selected data, we use pixel diffusion models as backbone; for Celeba-HQ and fMoW we use stable diffusion 1.5 as the backbone. For each target image, we sample eight $C_0$ from a uniform [0, 0.9] distribution. For each $C_0$, we sample 24 images. Then we compute the average $L^2$ distance between the sampled images and the target mean for each scale. We compute the R-squared coefficient ($R^2$) between the input perturbation scales and the normalized average residual norms (scale between 0-1). As shown in Table 4, we observe a very strong linearity in the above experiments. Fig. 1 also demonstrates linear semantic change visually. We provide additional analysis in the Appendix. In summary, the linearity widely exists for DDIM sampling regardless of dataset or model backbone, and it may heavily depend on the dataset distribution.

To further investigate how the linearity changes with the complexity of dataset, and different diffusion model backbones. We perform experiments to test the linearity of (1) pretrained diffusion models on a simple dataset such as FFHQ. (2) pretrained diffusion models on multimodal dataset (with many classes) such as ImageNet. (3) large pretrained foundation models such as Stable Diffusion Model (trained on complex multimodal dataset). We hypothesize that the linearity on out-of-distribution dataset will decreases, so we test the pretrained pixel-diffusion models on OOD datasets. While the training data for SD1.5 is very large (LAION-5B), we just test it on other multimodal datasets such as UCF-101 [39]

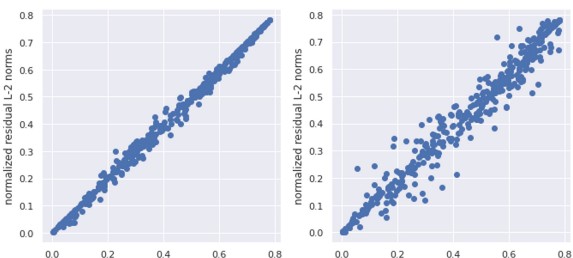

Figure 2: Quantitative demonstration of linearity when increasing scale of perturbation. With increased $\sin(C_0)$, the magnitude of perturbation increases, and the average $L^2$ distance between samples and the target image increases linearly. Left is the linearity on FFHQ dataset using pixel diffusion; Right is the linearity on Celeba-HQ dataset using Stable Diffusion 1.5.

and ImageNet [12]. Results show that the linearity decreases significantly when testing on OOD dataset for pixel-diffusion models. For foundation models, complexity of dataset for sampling does not affect the linearity significantly. The linearity decreases slightly comparing diffusion models trained on multimodal dataset to those trained on simple dataset. These results validate the analysis in the linear approximation error, and imply that a low probability density and a sudden change in the

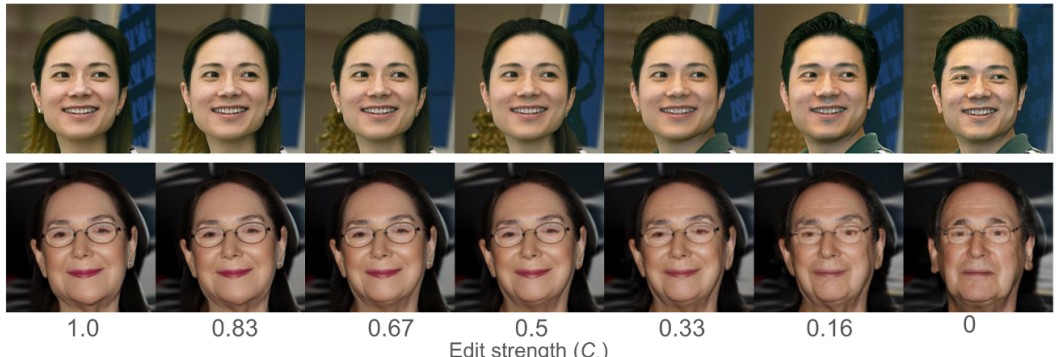

| 1.0 | 0.83 | 0.67 | 0.5 | 0.33 | 0.16 | 0 |

Edit strength ($C_0$)

Figure 3: Precise image editing with the proposed P-CCS algorithm. Source prompt: "A high-quality portrait of a man". Target prompt: "A high-quality portrait of a woman". The right-most images with edit strength as 0 are the source images.

data distribution may contribute to low linearity. We also find significant drop of sample quality with low linearity. Details and numbers can be found in the Appendix.

**Application 1: Precise Image Editing.** With the aid of linearity property, we can perform the application of precise image editing, by having the user enters a value of edit strength to precisely control the target image edited to that extent. The key idea is through the DDIM inversion to project both the source and target images back to the initial noise manifold. Specifically, we first compute $x_T^{(1)} = \text{DDIM}^{-1}(x_0, c_{\text{source}})$, $x_T^{(2)} = \text{DDIM}^{-1}(x_0, c_{\text{target}})$, and then perform spherical interpolation between $x_T^{(1)}$, and $x_T^{(2)}$ according to the user-specified editing strength. This is implemented using the proposed P-CCS algorithm with more details described in the Appendix (Alg. 4). As shown of two example images in Fig. 3, our algorithm can easily achieve a smooth and precise image editing guaranteed by the aforementioned linearity property.

## 5.2 Controllable Sampling

We propose CCS (Alg. 1) and P-CCS (Alg. 3) algorithms for controllable sampling close to a specified target image, constrained by a target MSE to the target mean. We validate that our algorithm can achieve this better than baselines while preserving good image quality. In addition, we demonstrate our algorithm's capability in generating personalized albums and improving sample quality.

**Application 2: Generating Personalized Album.** We perform experiments on generating personalized albums using both pixel diffusion models and latent diffusion models with our (P-)CCS algorithms. For benchmarking performance of different baselines, we propose a novel task of fixing MSE to a target image, and compare other metrics, which we call *controllable sampling*. The goal is to sample images as close to source as possible while keeping target diversity (MSE).

**Experimental setup:** We FFHQ-256 [22] and CelebA-HQ [43] test set images as target images. We use ADM (a pixel diffusion model) for FFHQ, and Stable Diffusion for CelebA-HQ. **Baselines:** We self-implement 5 baselines as comparison since no existing work is designed for the target task so some adaptation is necessary. Naive Linear interpolation with Controller (*LP-C*), Gaussian

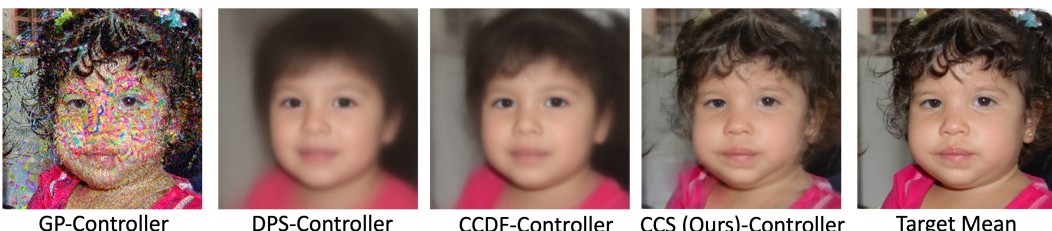

| GP-Controller | DPS-Controller | CCDF-Controller | CCS (Ours)-Controller | Target Mean |

Figure 4: We sample 120 images with a fixed target mean using different methods and analyze their sample mean (average pixel intensity). Our observations show that the sample mean of our method closely matches that of the original image.

Perturbation with Controller (*GP-C*), Diffusion Posterior Sampling [6] with Controller (*DPS-C*), ILVR with Controller (*ILVR-C*), Come-Closer-Diffuse-Faster with Controller (*CCDF-C* or *SDEdit-C*). **Metrics:** We adopt Peak Signal to Noise ratio (PSNR) for measuring whether sample mean is close to target

Table 2: Results of Pixel Diffusion models on the FFHQ Dataset with target rMSE set as 0.12.

| Method | PSNR↑ | SD↑ | CLIP-IQA↑ | MUSIQ↑ |
|---|---|---|---|---|
| GP-C | 18.88 | 0.028 | 0.701 | 45.88 |
| ILVR-C | 20.04 | 0.070 | 0.746 | 62.45 |
| DPS-C | 21.02 | 0.069 | 0.738 | 64.60 |
| CCDF-C | 23.52 | 0.088 | 0.746 | 66.15 |
| CCS (Ours)-C | **25.13** | **0.104** | **0.750** | **66.79** |

mean; LPIPS for measuring samples similarity to the source image; CLIP-IQA and MUSIQ for semantic/low-level image quality respectively; Standard Deviation (SD) for measuring sampling diversity. More experimental details and baseline implementation can be found in the Appendix.

**Results.** We observe that our CCS sampling method significantly outperforms all other methods in centering at a target mean constrained by a fixed rMSE distance, while surprisingly maintaining superior image perceptual quality and diversity.

Table 3: Results of the Stable Diffusion 1.5 on the CelebA-HQ dataset with target rMSE set as 0.07.

| Method | PSNR↑ | SD↑ | CLIP-IQA↑ | MUSIQ↑ |
|---|---|---|---|---|
| GP-C | 23.02 | 0.045 | 0.721 | 48.91 |
| LDPS-C | 24.56 | 0.034 | 0.721 | 29.07 |
| CCDF-C | 27.66 | 0.051 | **0.735** | 49.29 |
| CCS (Ours)-C | **30.29** | **0.053** | 0.732 | **49.66** |

Other posterior sampling methods such as DPS suffer from image quality degradation and diversity decrease, as shown by the quantitative results reported in Table. 2 and 3. Qualitatively, we observe that the sample means of other methods look blurry or noisy, as demonstrated in Fig. 4. More qualitative results can be found in the Appendix.

**Application 3: Improving Image Quality through P-CCS Sampling.** Note that in our (P-)CCS sampling algorithms, we perform spherical interpolation with a random Gaussian noise. Intuitively, if the initial noise is not Gaussian like falling in low-density probability region, conducting such interpolation will make it "more Gaussian" to increase the likelihood of that sample so as to enhance image qual-

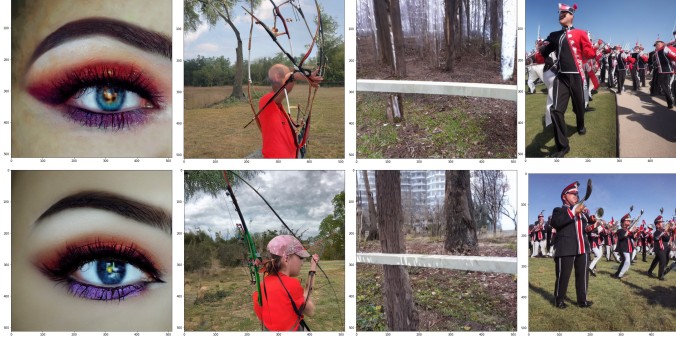

Figure 5: Top: Corrupted images with artifacts or unreasonable structures. Bottom: Improved images by P-CCS algorithm through spherical interpolation of initial noises.

ity. We provide a formal argument for this in the Appendix. Motivated by this, we propose to perform P-CCS *at some timesteps* of reverse DDIM sampling. As shown in Fig. 5, we observe that the sample quality can be improved significantly by this simple method, which supports the potential of our findings to introduce a new post-training mechanism for enhancing image generation. More experiment details and quantitative results are described in the Appendix.

## 6  Conclusion

In this work, we unveil an interesting linear response to perturbation phenomenon both theoretically and empirically in diffusion models. we also study a new problem: how to sample images with a target mean and target MSE. We present a novel sampling algorithm along with a new controller method for achieving this goal. Extensive experiments show that our proposed method samples the closest to the target mean when controlling the MSE compared to other methods, while maintaining superior image quality and diversity. The limitations of our work include: (1) Controlling other interesting statistical properties beyond sample mean with MSE is left as future work. (2) There might be some artifact samples that exhibit overlapping patterns. (3) DDIM inversion may not be perfectly standard Gaussian, which may hurt sample quality. We believe the linearity property will be important for designing better latent space for large-scale diffusion models.

## Acknowledgments

LS acknowledges funding supports from National Science Foundation (NSF) (IIS-2435746), Defense Advanced Research Projects Agency (DARPA), Hyundai America Technical Center, Inc. (HATCI), University of Michigan MICDE Catalyst Grant Award and MIDAS PODS Grant Award. GW acknowledges support from the National Science Foundation (NSF) (DMS-2210849) and the Adobe Data Science Award.

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

# A Proofs

## A.1 Proof in Section 3.2

*Proof of Proposition 1.* Let $L_t(\mathbf{x}) := \eta_t \mathbf{x} + \lambda_t \nabla_{\mathbf{x}_t} \log p_t(\mathbf{x})$ be the one-step recursion. Our $\mathbf{x}_0(a, t)$ is formally defined as $L_1 \circ L_2 \circ \ldots \circ L_T(a)$.

The second-order differentiability of $p_t$ implies the score function $\nabla \log p_t$ is first-order differentiable. Let $H_t$ be the Hessian matrix of $\log p_t$ ($H_t^{i,j} := \partial^2 \log p_t / \partial_i \partial_j$). We have

$$\nabla \log p_t(\mathbf{x}) = \nabla \log p_t(\mathbf{w}) + H_t(\mathbf{w})(\mathbf{x} - \mathbf{w}) + o(\|\mathbf{x} - \mathbf{w}\|_2).$$

Therefore, for any fixed direction $\mathbf{w}$ of unit length and $\delta \in \mathbb{R}$,

$$\begin{aligned}
L_T(\mathbf{x} + \delta\mathbf{w}) &= \eta_T(\mathbf{x} + \delta\mathbf{w}) + \lambda_T \nabla_{\mathbf{x}} \log p_T(\mathbf{x} + \delta\mathbf{w}) \\
&= \eta_T \mathbf{x} + \lambda_T \nabla_{\mathbf{x}} \log p_T(\mathbf{x}) + \lambda_T \delta H_T(\mathbf{x})\mathbf{w} + \delta\eta_T\mathbf{w} + o(\delta) \\
&= L_T(\mathbf{x}) + \delta(\eta_T + \lambda_T H_T(\mathbf{x}))\mathbf{w} + o(\delta) \\
&= L_T(\mathbf{x}) + \delta\gamma_T(\mathbf{x})\mathbf{w} + o(\delta)
\end{aligned}$$

where $\gamma_T(\mathbf{x})$ is defined as

$$\gamma_T(\mathbf{x}) = \eta_T + \lambda_T H_T(\mathbf{x}),$$

is a matrix-valued function which is bounded if the norm of the Hessian of $\log p_t$ is bounded.

Applying $L_{T-1}$ on both sides of the above formula:

$$\begin{aligned}
L_{T-1} \circ L_T(\mathbf{x} + \delta\mathbf{w}) &= L_{T-1} \circ (L_T(\mathbf{x}) + \delta\gamma_T(\mathbf{x})\mathbf{w} + o(\delta)) \\
&= \eta_{T-1} L_T(\mathbf{x}) + \delta\eta_{T-1}\gamma_T(\mathbf{x})\mathbf{w} + o(\delta) + \lambda_{T-1} \nabla \log p_{T-1}\left( L_T(\mathbf{x}) + \delta\gamma_T(\mathbf{x})\mathbf{w} + o(\delta) \right) \\
&= \underbrace{\eta_{T-1} L_T(\mathbf{x}) + \lambda_{T-1} \nabla \log p_{T-1}(L_T(\mathbf{x}))}_{\text{recursion on the unperturbed data } \mathbf{x}} + \underbrace{\delta\eta_{T-1}\gamma_T(\mathbf{x})\mathbf{w} + \delta\lambda_{T-1} H_{T-1}(L_T(\mathbf{x}))\gamma_T(\mathbf{x})\mathbf{w}}_{\text{linear term}} \\
&\quad + \underbrace{o(\delta)}_{\text{lower order term}} \\
&= L_{T-1} \circ L_T(\mathbf{x}) + \delta\gamma_{T-1}(\mathbf{x})\mathbf{w} + o(\delta).
\end{aligned}$$

where

$$\gamma_{T-1}(\mathbf{x}) := (\eta_{T-1} I + \lambda_{T-1} H_{T-1}(L_T(\mathbf{x}))) \gamma_T(\mathbf{x})$$

So we have

$$\mathbf{x}_0(\mathbf{x} + \delta\mathbf{w}, T) := L_0 \circ \cdots L_{T-1} \circ L_T(\mathbf{x}) + \delta\gamma_0(\mathbf{x})\mathbf{w} + o(\delta)$$

Now let $\lambda$ be the scale of the perturbation, such that $\lambda > 0$ and $\lambda \in \mathbb{R}$, and let $\Delta\mathbf{x}$ be the unit-length perturbation to the initial noise $\mathbf{x}_T$, we have:

$$\mathbf{x}_0(\mathbf{x}_T + \lambda\Delta\mathbf{x}, T) = \mathbf{x}_0(\mathbf{x}_T, T) + \lambda\gamma_0(\mathbf{x}_T)\Delta\mathbf{x} + o(\lambda)$$

We could continue applying $L_{T-2}, L_{T-3} \ldots, L_1$ on the above formula, and conclude:

$$\mathbf{x}_0(\mathbf{x}_T + \lambda\Delta\mathbf{x}, T) = x_0(\mathbf{x}_T) + \lambda\gamma_0(\mathbf{x}_T)\Delta\mathbf{x} + o(\lambda). \tag{5}$$

We might be particularly interested in the distance $\|\mathbf{x}_0(\mathbf{x}_T + \lambda\Delta\mathbf{x}, T) - \mathbf{x}_0(\mathbf{x}_T, T)\|$, our calculation directly implies:

$$\|\mathbf{x}_0(\mathbf{x}_T + \lambda\Delta\mathbf{x}, T) - \mathbf{x}_0(\mathbf{x}_T, T)\|_2 = $$
$$\|\lambda\gamma_0(\mathbf{x}_T)\Delta\mathbf{x}\|_2 + o(\lambda) = \lambda\|\gamma_0(\mathbf{x}_T)\Delta\mathbf{x}\|_2 + o(\lambda). \tag{6}$$

by applying triangle inequality twice:

$$\|\lambda\gamma_0(\mathbf{x}_T)\Delta\mathbf{x}\|_2 - \|o(\lambda)\|_2 \leq \|\mathbf{x}_0(\mathbf{x}_T + \lambda\Delta\mathbf{x}, T) - \mathbf{x}_0(\mathbf{x}_T)\|_2 \leq \|\lambda\gamma_0(\mathbf{x}_T)\Delta\mathbf{x}\|_2 + \|o(\lambda)\|_2$$

$\square$

## A.2 Proof in Section 3.3

We first state the detailed assumptions posed in Section 3.3. Define the function

$$h(t, \mathbf{y}) := -\frac{1}{2}\sqrt{\alpha(t)}\frac{\alpha'(t)}{\alpha_t^2}\nabla \log p_t\left(\frac{\mathbf{y}}{\sqrt{\sigma^2(t)+1}}\right).$$

We assume this function has a continuous derivative (i.e., $C^1$) on the whole space $[0, T] \times \mathbb{R}^m$. Moreover, we assume there exists $C(t)$ such that:

$$\|h(t, \mathbf{y}) - h(t, \mathbf{x})\|_2 \le C(t)\|\mathbf{y} - \mathbf{x}\|_2,$$

for every $\mathbf{x}, \mathbf{y}, t$, and $\max_{t \in [0,T]} C(t) \le C < \infty$.

*Proof of Proposition 2.* We first show the ODE (4) exists a unique solution. We can rewrite the (4) as:

$$\begin{aligned}
\mathrm{d}\bar{\mathbf{x}}_t &= -\sqrt{1-\alpha(t)}\nabla \log p_t\left(\frac{\bar{\mathbf{x}}_t}{\sqrt{\sigma^2(t)+1}}\right)\mathrm{d}\sigma(t) \\
&= -\sigma'(t)\sqrt{1-\alpha(t)}\nabla \log p_t\left(\frac{\bar{\mathbf{x}}_t}{\sqrt{\sigma^2(t)+1}}\right)\mathrm{d}t \\
&= -\frac{1}{2}\sqrt{\alpha(t)}\frac{\alpha'(t)}{\alpha_t^2}\nabla \log p_t\left(\frac{\bar{\mathbf{x}}_t}{\sqrt{\sigma^2(t)+1}}\right)\mathrm{d}t \quad \text{as } \sigma(t) = \sqrt{(1-\alpha(t))/\alpha(t)} \\
&= h(t, \bar{\mathbf{x}}_t)\mathrm{d}t.
\end{aligned}$$

Given $h(t, \mathbf{y}) \in C^1$ and unifomrly Lipschitz in $\mathbf{y}$, it follows from the Picard-Lindelöf Theorem (e.g., Theorem 1.1 of [19]) that our ODE (4) has a unique solution for any initialization $\bar{\mathbf{x}}_T = \bar{\mathbf{x}}$.

Next, it follows from Theorem 3.1 of [19] that the solution $\bar{\mathbf{x}}_0(\bar{\mathbf{x}}, T) \in C^1$, i.e., the solution depends continuously and differentiably on its initialization $\bar{\mathbf{x}}$. Thus,

$$\bar{\mathbf{x}}_0(\mathbf{x}_T + \lambda\Delta\mathbf{x}, T) = \bar{\mathbf{x}}_0(\mathbf{x}_T, T) + \lambda J_{\bar{\mathbf{x}}}(\mathbf{x}_T)\Delta\mathbf{x} + o(\lambda),$$

where $J_{\mathbf{x}}$ is the Jabobian matrix of the function $\bar{\mathbf{x}}_0(\bar{\mathbf{x}}, t)$ with respect to $\bar{\mathbf{x}}$. This concludes the proof of Proposition 2. $\square$

*Proof of Proposition 3.* Let $\bar{\mathbf{x}}_T$ and $\bar{\mathbf{x}}_T + \lambda\Delta\mathbf{x}$ be two fixed initializations. Define

$$\mathbf{y}_t := \bar{\mathbf{x}}_t(\bar{\mathbf{x}}_T) - \bar{\mathbf{x}}_t(\bar{\mathbf{x}}_T + \lambda\Delta\mathbf{x})$$

as the difference between the solutions of (4) at time $t \in [0, T]$.

Taking derivative on $\mathbf{y}$ with respect to $t$ yields:

$$\mathbf{y}'_t = h(t, \bar{\mathbf{x}}_t(\bar{\mathbf{x}}_T)) - h(t, \bar{\mathbf{x}}_t(\bar{\mathbf{x}}_T + \lambda\Delta\mathbf{x})).$$

By the Lipschitz continuity:

$$\|\mathbf{y}'_t\|_2 \le C\|\bar{\mathbf{x}}_t(\bar{\mathbf{x}}_T) - \bar{\mathbf{x}}_t(\bar{\mathbf{x}}_T + \lambda\Delta\mathbf{x})\|_2 = C(t)\|\mathbf{y}_t\|_2$$

Denote $\mathbf{y}_t$ by $(\mathbf{y}_{1,t}, \mathbf{y}_{2,t}, \ldots, \mathbf{y}_{m,t})^\top$, we have:

$$\begin{aligned}
\frac{\mathrm{d}\|\mathbf{y}_t\|_2}{\mathrm{d}t} &= \frac{\mathrm{d}\sqrt{\sum_{i=1}^m \mathbf{y}_{i,t}^2}}{\mathrm{d}t} \\
&= \frac{1}{2}\frac{\sum_{i=1}^m 2\mathbf{y}_{i,t}\mathbf{y}'_{i,t}}{\sqrt{\sum_{i=1}^m \mathbf{y}_{i,t}^2}} \\
&= \frac{\sum_{i=1}^m \mathbf{y}_{i,t}\mathbf{y}'_{i,t}}{\|\mathbf{y}_t\|_2} \\
&\le \frac{\|\mathbf{y}_t\|_2\|\mathbf{y}'_t\|_2}{\|\mathbf{y}_t\|_2} \quad \text{Cauchy-Schwarz inequality} \\
&= \|\mathbf{y}'_t\|_2.
\end{aligned}$$

Thererfore, we have

$$\frac{\mathrm{d}\|\mathbf{y}_t\|_2}{\mathrm{d}t} \leq C(t)\|\mathbf{y}_t\|_2.$$

Applying Grönwall's inequality on the function $\|\mathbf{y}_t\|$, we have:

$$\|\mathbf{y}_t\|_2 \leq \exp\left(\int_t^T C(t)\mathrm{d}t\right)\|\lambda\Delta\mathbf{x}\|_2$$

for every $0 \leq t \leq T$. Taking $t = 0$, we have

$$\|\bar{\mathbf{x}}_0(\mathbf{x}_T + \lambda\Delta\mathbf{x}, T) - \bar{\mathbf{x}}_0(\mathbf{x}_T, T)\|_2 \leq \exp\left(\int_0^T C(t)\mathrm{d}t\right)|\lambda|\|\Delta\mathbf{x}\|_2.$$

as claimed in Proposition 3. $\qquad\square$

### A.3 Proof in Section 4

*Proof of Proposition 5.* It is known

$$\begin{aligned}
\mathbb{E}[\|\mathbf{x} + \Delta\mathbf{x}\|_2^2] &= \|\mathbb{E}[\mathbf{x} + \Delta\mathbf{x}]\|_2^2 + \mathrm{tr}(\mathrm{Cov}[\mathbf{x} + \Delta\mathbf{x}]) \\
&= \|\mathbf{x}\|_2^2 + \mathrm{tr}(\mathrm{Cov}[\Delta\mathbf{x}]) \\
&\geq \|\mathbf{x}\|_2^2
\end{aligned}$$

The equality is taken if and only if $\mathrm{tr}(\mathrm{Cov}[\Delta\mathbf{x}]) = \sum_i \mathrm{Var}[\Delta\mathbf{x}_i] = 0$. This is equivalent to saying that all components of $\Delta\mathbf{x}$ are deterministic. Therefore, almost surely, $\Delta\mathbf{x} = \mathbb{E}[\Delta\mathbf{x}] = 0$. $\quad\square$

*Proof of Proposition 6.* Given a standard normal vector $\epsilon$, we claim the following holds:

$$\frac{\|\epsilon - \mathbf{x}_T\|_2^2}{d} = \frac{\|\epsilon\|_2^2}{d} + \frac{\|\mathbf{x}_T\|_2^2}{d} + \frac{-2\epsilon \cdot \mathbf{x}_T}{d} \to 2$$

in $L^2$ as $d \to \infty$. To see this, notice the first term is

$$\frac{\sum_{i=1}^d \epsilon_i^2}{d}$$

which converges to 1 by the law of large numbers, since $\mathbb{E}[\epsilon_i^2] = 1$. The second term converges to 1 by our assumption. The last term converges to 0 in $L^2$ as

$$\mathbb{E}\left[\left\|\frac{-2\epsilon \cdot \mathbf{x}_T}{d}\right\|^2\right] = \frac{4\mathbb{E}[\sum_i \mathbf{x}_{T,i}^2 \mathbb{E}[\epsilon_i^2]]}{d^2} = \frac{4(d + o(d))}{d^2} \to 0.$$

Therefore the distance $\|\epsilon - \mathbf{x}_T\|_2$ converges to $2\sqrt{d}$ as $d \to \infty$. Similarly we can show $\theta(\epsilon, \mathbf{x}_T)$, the angle between $\epsilon$ and $\mathbf{x}_T$ converges to $\pi/2$ as $d \to \infty$. In other words, $\epsilon$ is approximately orthogonal to $\mathbf{x}_T$ when the dimension $d$ is large.

Therefore, with probability $1 - o(1)$, the angle $\theta$ in Algorithm 1 is $\pi/2 \pm o(1)$, and $\|\epsilon - \mathbf{x}_T\|_2/2\sqrt{d} = 1 \pm o(1)$ as $d \to \infty$. Fix any $M \leq (2 - \delta)\sqrt{d}$, since the spherical interpolation smoothly interpolate between $\mathbf{x}_0$ and $\epsilon$, there exists a $C$ satisfying Algorithm 1 with input $C$ output $\mathbf{x}_T'$ with distance $M$ to $\mathbf{x}_T$ with probability $1 - o(1)$.

We can indeed find an explicit $C_0$ with slightly weaker guarantees, set

$$C_0 = \cos^{-1}\left(1 - \frac{M^2}{2\|\mathbf{x}_t\|_2^2}\right).$$

Then with probability $1 - o(1)$, $C_0 \in (0, \pi/2)$, and

$$\left\|\frac{\sin(C_0)}{\sin(\theta)} \cdot \epsilon + \frac{\sin(\theta - C_0)}{\sin(\theta)} \cdot \mathbf{x}_T - \mathbf{x}_T\right\| \leq \left\|\frac{\sin(C_0)}{\sin(\theta)} \cdot \epsilon + \frac{\sin(\theta - C_0)}{\sin(\theta)} \cdot \mathbf{x}_T - \sin(C_0)\epsilon - \sin(\theta - C_0)\mathbf{x}_T\right\|$$

$$+ \|\sin(C_0)\epsilon + \sin(\theta - C_0)\mathbf{x}_T - \mathbf{x}_T\|$$

by triangle's inequality. Meanwhile, the first term is $o(\sqrt{d})$ as $\sin(\theta) = \sin(\pi/2 + o(1)) = 1 + o(1)$ and $\cos(\theta) = 1 - o(1)$. The square of the second term is

$$\|\sin(C_0)\epsilon + \sin(\theta - C_0)\mathbf{x}_T - \mathbf{x}_T\|^2 = \sin(C_0)^2\|\epsilon\|^2 + (1 - \sin(\theta - C_0))^2\|\mathbf{x}_T\|^2 + 2\sin(C_0)(\sin(\theta - C_0) - 1)\epsilon \cdot \mathbf{x}_T$$
$$= \sin(C_0)^2(d + o(1)) + (1 - \sin(\theta - C_0))^2(d + o(1)) + o(d)$$

The last term is $o(d)$ as $\epsilon \cdot \mathbf{x}_T/d \to 0$ as we analyzed above. Using again $\theta = \pi/2 + o(1)$, we know $\sin(\theta - C_0) = \sin(\pi/2 - C_0) + o(1) = \cos(C_0) + o(1)$. Hence we clean the above equation:

$$\|\sin(C_0)\epsilon + \sin(\theta - C_0)\mathbf{x}_T - \mathbf{x}_T\|^2 = d(\sin(C_0)^2 + (1 - \cos(C_0))^2) + o(d)$$
$$= d(2 - 2\cos(C_0)) + o(d)$$
$$= d\left(2 - 2 + \frac{M^2}{\|\mathbf{x}_t\|_2^2}\right) + o(d)$$
$$= M^2 + o(d),$$

where the last equality follows from $\|\mathbf{x}_T\|_2^2 = d + o(1)$. Finally, taking the square root and plugging back into the triangle inequality, we have:

$$\|\mathbf{x}_T' - \mathbf{x}_T\| = M + o(\sqrt{d}).$$

$\square$

*Proof of Proposition 7.* We can write $\mathbf{x} = r_0\mathbf{x}_0$ where $r_0 = \|\mathbf{x}\|$ and $\mathbf{x}_0 = \mathbf{x}/r_0$ belongs to $S^{d-1}$, the unit sphere in $\mathbb{R}^d$.

Meanwhile, it is well known that we can generate $\epsilon \sim \mathbb{N}(0, I_d)$ via 1) sample $r_1^2 \sim \chi^2(d)$ from the chi-squared distribution with parameter $d$, 2) sample $\mathbf{u} \sim \text{Unif}(S^{d-1})$ uniformly on the $d$-dim unit sphere, 3) set $\epsilon = r_1 u$. Therefore, let

$$\theta = \theta(\mathbf{u}) := \arccos\langle\mathbf{x}_0, \mathbf{u}\rangle$$
$$s(\theta) := \frac{\sin(c\theta)}{\sin\theta}$$
$$t(\theta) := \frac{\sin((1-c)\theta)}{\sin\theta}.$$

We rewrite $\mathbf{y}$ as

$$\mathbf{y} = s(\theta)r_1\mathbf{u} + t(\theta)r_0\mathbf{x}_0$$

Now we calculate $\mathbb{E}[\|\mathbf{y}\|^2]$:

$$\mathbb{E}[\|\mathbf{y}\|^2] = \mathbb{E}[s(\theta)^2 r_1^2\|\mathbf{u}\|^2] + \mathbb{E}[t(\theta)^2 r_0^2\|\mathbf{x}_0\|^2] + \mathbb{E}[2r_0 r_1 s(\theta)t(\theta)\langle\mathbf{u}, \mathbf{x}_0\rangle]$$

Our first claim is the cross term $\mathbb{E}[2r_0 r_1 s(\theta)t(\theta)\langle\mathbf{u}, \mathbf{x}_0\rangle]$ is zero. To see this, we first observe since $r_1$ and $\mathbf{u}$ are independent, we have

$$\mathbb{E}[2r_0 r_1 s(\theta)t(\theta)\langle\mathbf{u}, \mathbf{x}_0\rangle] = 2r_0\mathbb{E}[r_1]\mathbb{E}[s(\theta)t(\theta)\langle\mathbf{u}, \mathbf{x}_0\rangle]$$

We examine the expectation $\mathbb{E}[s(\theta)t(\theta)\langle\mathbf{u}, \mathbf{x}_0\rangle]$ where the only random variable is the direction vector $\mathbf{u}$. Replacing $\mathbf{u}$ with $-\mathbf{u}$ sends the angle $\theta$ to $\pi - \theta$, which will not change the value of $s(\theta), t(\theta)$. However, the inner product flips sign: $\langle-\mathbf{u}, \mathbf{x}_0\rangle = -\langle\mathbf{u}, \mathbf{x}_0\rangle$. Consequently, the function of interest:

$$F(\mathbf{u}) := s(\theta(\mathbf{u}))t(\theta(\mathbf{u}))\langle u, \mathbf{x}_0\rangle$$

is an odd function of $\mathbf{u}$. Therefore,

$$\mathbb{E}_{\mathbf{u}\sim\text{Unif}(S^{d-1})}[s(\theta)t(\theta)\langle\mathbf{u}, \mathbf{x}_0\rangle] = 0.$$

Now the expected distance simplifies to

$$\mathbb{E}[\|\mathbf{y}\|^2] = \mathbb{E}[s(\theta)^2 r_1^2 \|\mathbf{u}\|^2] + \mathbb{E}[t(\theta)^2 r_0^2 \|\mathbf{x}_0\|^2]$$
$$= d\mathbb{E}[s(\theta)^2] + r_0^2 \mathbb{E}[t(\theta)^2]$$

The next observation is the following trigonometric identity:

$$s(\theta)^2 + t(\theta)^2 + 2s(\theta)t(\theta)\cos(\theta) = 1$$

for any $c \in (0,1), \theta \in (0,\pi)$.

To show this identity, set $\alpha = c\theta, \beta = (1-c)\theta$. Then $s = \sin(\alpha)/\sin(\alpha+\beta), t = \sin(\beta)/\sin(\alpha+\beta)$. The left hand side of the claimed identity equals

$$\frac{\sin^2(\alpha) + \sin^2(\beta) + 2\sin(\alpha)\sin(\beta)\cos(\alpha+\beta)}{\sin^2(\alpha+\beta)}$$

Now we expand $\sin^2(\alpha+\beta)$ as

$$\sin^2(\alpha+\beta) = (\sin(\alpha)\cos(\beta) + \cos(\alpha)\sin(\beta))^2$$
$$= \sin^2(\alpha)\cos^2(\beta) + \cos^2(\alpha)\sin^2(\beta) + 2\sin(\alpha)\cos(\alpha)\sin(\beta)\cos(\beta)$$
$$= \sin^2(\alpha) - \sin^2(\alpha)\sin^2(\beta) + \sin^2(\beta) - \sin^2(\alpha)\sin^2(\beta) + 2\sin(\alpha)\cos(\alpha)\sin(\beta)\cos(\beta)$$
$$= \sin^2(\alpha) + \sin^2(\beta) + 2\sin(\alpha)\sin(\beta)(\cos(\alpha)cos(\beta) - \sin(\alpha)\sin(\beta))$$
$$= \sin^2(\alpha) + \sin^2(\beta) + 2\sin(\alpha)\sin(\beta)\cos(\alpha+\beta),$$

as claimed.

Leveraging this equality $s(\theta)^2 + t(\theta)^2 + 2s(\theta)t(\theta)\cos(\theta) = 1$ and taking expectation with respect to $\mathbf{u}$ implies:

$$\mathbb{E}[s(\theta)^2] + \mathbb{E}[t(\theta)^2] = 1,$$

as the cross term is zero as proved before.

Now let $s_c := \mathbb{E}[s(\theta)^2] \in (0,1)$, we have

$$\mathbb{E}[\|\mathbf{y}\|^2] = ds_c + r_0^2(1 - s_c)$$

Therefore

$$\delta(\mathbf{y}) = |\mathbb{E}[\|\mathbf{y}\|^2] - d| = (1 - s_c)|d - r_0^2| = (1 - s_c)|\|\mathbf{x}\|^2 - d| \le |\|\mathbf{x}\|^2 - d| = \delta(\mathbf{x})$$

$\square$

## A.4 Why linear interpolation fails

To further support this point, we provide additional theoretical analysis to justify why simple linear interpolation does not work well for sampling. Formally, let $\|\Delta x\|_2 = \|x\|_2$, and its direction is uniformly distributed, and $\mathbb{E}[\Delta x] = 0$. For $0 < \alpha < 1$, we have:

$$\mathbb{E}\left[\|\alpha x + (1 - \alpha)\Delta x\|_2^2\right] < \|x\|_2^2$$

**Proof.** We have:

$$\|\alpha x + (1 - \alpha)\Delta x\|_2^2 = \alpha^2 \|x\|_2^2 + 2\alpha(1 - \alpha)(x \cdot \Delta x) + (1 - \alpha)^2 \|\Delta x\|_2^2$$

Since $\mathbb{E}\left[\alpha(1 - \alpha)(x \cdot \Delta x)\right] = \alpha(1 - \alpha)x \cdot \mathbb{E}[\Delta x] = 0$, we have:

$$\mathbb{E}\left[\|\alpha x + (1 - \alpha)\Delta x\|_2^2\right] = \alpha^2 \|x\|_2^2 + (1 - \alpha)^2 \|\Delta x\|_2^2 = (\alpha^2 + (1 - \alpha)^2)\|x\|_2^2$$

Since $2\alpha(\alpha - 1) < 0$, we have $(\alpha^2 + (1 - \alpha)^2) < 1$, and we can conclude that using simple linear interpolation cannot preserve the norm, leading to falling apart from the Gaussian sphere.

# B More Analysis on Linearity

## B.1 Validation of Linearity Phenomenon on changing $C_0$

**Experimental setting.** We perform extensive experiments on both pixel diffusion models on the FFHQ and CIFAR-10 dataset and latent diffusion models on the Celeba-HQ and fMoW dataset. For each experiment, we first sample 50 images as target images from each validation dataset from FFHQ [22], CIFAR-10[25], and Celeba-HQ [43]. We also pick one images each class from the validation set of the fMoW dataset [5] for further verification. Then for the FFHQ and CIFAR-10 selected data, we use pixel diffusion models as backbone; for Celeba-HQ and fMoW we use stable diffusion 1.5 as the backbone. The prompt for Celeba-HQ is given by "A high quality photo of a face" and the prompt for fMoW is given by "satellite images". Then, we use each image as a target mean and perform CCS sampling as in Alg.1.

For each target image, we sample eight $C_0$ from a uniform [0, 0.9] distribution. For each $C_0$, we sample 24 images. Then we compute the average $L^2$ distance between the sampled images and the target mean for each scale.

**Evaluations.** To quantitatively evaluate the linearity phenomenon, we compute the R-square between the input perturbation scales and the normalized average residual norms (scale between 0-1) for 4 datasets with both pixel diffusion models and latent diffusion models. Note that since different target means can lead to different slopes by different Hessian matrices, we normalize the residual norms. Specifically, we compute empirical slope $a$ and bias $b$ between $x = \sin(C_0)$ and $y = \mathbb{E}[\|\mathbf{x}_0' - \mathbf{x}_0\|_2]$ each target mean, and then normalize the average $L^2$ distance to be: $y' = \frac{y-b}{a}$.

**Results.** We observe a very strong linearity in the above experiments. Especially for pixel diffusion models, the R-square exceeds 0.98 for both datasets, which indicates almost a perfect linear relationship. For latent diffusion models, the linearity is slightly weaker, but still above 0.94 in R-square for both datasets. This is expected since Stable Diffusion use a nonlinear autoencoder and trained on a different dataset. We also present more quantitative results in Fig. 2 and qualitative results in Fig. 1. Surprisingly, we also observe a very linear semantic change in additional to pixel-value change.

| Pixel Diffusion Models | | Latent Diffusion Models | |
|---|---|---|---|
| FFHQ | CIFAR-10 | CelebA-HQ | fMoW |
| 0.995 | 0.988 | 0.959 | 0.947 |

Table 4: R-square between scales of input perturbation and normalized residual norms

## B.2 Validation of Compositional Linearity

**Experimental setting.** Here we just have motivation similar to the previous one. We first sample a random noise $\epsilon_0 \in N(0, I)$, and then another random noise $\epsilon_1 \in N(0, I)$. We perform spherical interpolation on $\epsilon_0$ and $\epsilon_1$, and inference using a joint noise. We can adjust $C_0$ to decrease the strength of sspherical interpolation to preserve orginal image structure. This. We sample $C_0$ uniformly and continue sample $\epsilon_0$ and $\epsilon_1$ for evaluation. Then we compare the cosine distance between DDIM($slerp(\epsilon_0, \epsilon_1, C_0)$) and $a$DDIM($\epsilon_0$) + $b$DDIM($\epsilon_1$), where $a$ and $b$ are given by $slerp(\epsilon_0, \epsilon_1, C_0)$. We still find very large linearity, and the linearity decreases for Latent Diffusion Models. The quantitative results using four datasets, each with 100 evaluations are attached below in Table 5:

| Pixel Diffusion Models | | Latent Diffusion Models | |
|---|---|---|---|
| FFHQ | CIFAR-10 | CelebA-HQ | fMoW |
| 0.958 | 0.942 | 0.901 | 0.920 |

Table 5: Cosine similarity between samples generated by spherical interpolated initial noise, and linear combination of samples.

## B.3 Mathematical Explanation of the Linearity

**Bound the Linear Approximation Error of Score Function**

Let

$$z = \sqrt{\delta}x_0 + \sqrt{1-\delta}\,\epsilon, \quad f(x) = \text{data distribution}$$

$$\nu_\delta(x) = f\left(\frac{x}{\sqrt{\delta}}\right) \cdot \frac{1}{\sqrt{\delta}}, \quad u(x) = \mathcal{N}(0, 1-\delta)$$

Then the density of $z$ is:

$$f_z(z) = (\nu * u)(z)$$

**Score Function and Its Derivatives**

The score function is:

$$\nabla \log f_z(z) = \frac{\mathbb{E}_{u(x)}[\nabla\nu(z+x)]}{\mathbb{E}_{u(x)}[\nu(z+x)]}$$

Let:

$$A(z) := \mathbb{E}_{u(x)}[\nu(z+x)], \quad B(z) := \mathbb{E}_{u(x)}[\nabla\nu(z+x)], \quad C(z) := \mathbb{E}_{u(x)}[\nabla^2\nu(z+x)]$$

Then the second derivative is:

$$\nabla^2 \log f_z(z) = \frac{C(z)}{A(z)} - \frac{B(z)B(z)^\top}{A(z)^2}$$

Assuming $\nabla A(z) = B(z)$, $\nabla B(z) = C(z)$, and $\nabla C(z)$ exists, the third derivative becomes:

$$\nabla^3 \log f_z(z) = \frac{\nabla C(z)}{A(z)} - \frac{C(z) \otimes B(z)}{A(z)^2}$$
$$- \frac{C(z)B(z)^\top + B(z) \otimes C(z)}{A(z)^2} + \frac{2B(z)B(z)^\top \otimes B(z)}{A(z)^3}$$

**Norm Bound on the Third Derivative**

Suppose:

$$\|C(z)\|_2 \le c, \quad \|B(z)\|_2 \le b, \quad A(z) \ge a > 0, \quad \|\nabla C(z)\|_2 \le d$$

Then:

$$\left\|\nabla^3 \log f_z(z)\right\|_2 \le \frac{d}{a} + \frac{3bc}{a^2} + \frac{2b^3}{a^3}$$

**Taylor Approximation and Linearization Error Bound**

Let $z_0$ be a reference point. Then the second-order Taylor expansion of the score function is:

$$\nabla \log f_z(z) \approx \nabla \log f_z(z_0) + \nabla^2 \log f_z(z_0)(z - z_0)$$

The remainder (linearization error) satisfies:

$$\|\nabla \log f_z(z) - \nabla \log f_z(z_0) - \nabla^2 \log f_z(z_0)(z-z_0)\|_2 \le \frac{1}{2} \sup_{t \in [0,1]} \left\|\nabla^3 \log f_z(z_0 + t(z-z_0))\right\|_2 \cdot \|z-z_0\|_2^2$$

Using the third derivative bound, we get the formula for linear approximation error bound:

$$\|\mathcal{E}(z, z_0)\|_2 \le \frac{1}{2}\left(\frac{d}{a} + \frac{3bc}{a^2} + \frac{2b^3}{a^3}\right) \cdot \|z - z_0\|_2^2$$

By this bound, we may argue that the probability density of the sampling center, the curvature and gradient of the distribution (smoothness) impact the linearity error the most. If we are in the high probability region (a local maximum), assume that the curvature of probability distribution at that place is small, we will have low linear approximation error. So more complicated dataset may have less linearity since it is more likely to have discontinuous regions or low-density regions. This mathematical derivation explains the decrease of linearity from Pixel diffusion models to LDMs as shown in Table: 5.

## B.4 Linearity Analysis for OOD and Multimodal data

To further investigate this relationship, we perform additional experiments on three more datasets to test how the linearity changes for multimodal distributions. We are interested in:

1. How is the linearity when the model is trained on a multimodal dataset and samples an out-of-distribution target image from another highly multimodal dataset, or an OOD simple dataset?

2. How is the linearity when the model is trained on a simple dataset (for example, FFHQ) and samples on an out-of-distribution multimodal dataset, or an OOD simple dataset?

3. How is the linearity when training on a simple dataset and sampling on the same dataset?

4. How is the linearity comparing model trained on multimodal dataset in-distribution and model trained on simple dataset testing in-distribution?

In our experiment section of the paper, we cover partially 1, 3, and 4. We test pixel diffusion models on FFHQ and CIFAR datasets. We also test the Stable Diffusion model (1.5) trained on a complex dataset (LAION-5B) on the human face dataset (CelebA-HQ). We observe that diffusion models trained on simple datasets (FFHQ) and tested on distributions exhibiting significant differences show stronger linearity than Stable Diffusion trained on CelebA-HQ.

We observed in our paper that linearity decreases slightly when comparing diffusion models trained on multimodal datasets to diffusion models with simple training data. CIFAR-10, being a dataset with different classes, shows a slightly lower linearity score compared to FFHQ even though CIFAR-10 is lower resolution. Empirically, we observe sudden changes of image semantics occasionally.

1. We pick 4 images from classes 0, 4, . . . , 99 from the validation set of ImageNet.

2. We pick five videos from classes: "Applying Eye Makeup", "Baby Crawling", "Billiard", and "Blow Dry Hair" from the UCF-101 dataset, and sample five frames per video.

3. We pick 10 images from each organ site in the AAPM dataset, which consists of CT scans of different body parts. These datasets are different from the training data of SD1.5.

We use the same linearity testing methods as in Section 5.1 of our main paper (and also in Appendix B). We first summarize the results for Stable Diffusion 1.5:

| Dataset | ImageNet | UCF-101 | CelebA-HQ | fMoW |
|---|---|---|---|---|
| $R^2$ | 0.960 | 0.962 | 0.959 | 0.947 |
| Cosine similarity of linear combinations | 0.922 | 0.924 | 0.901 | 0.920 |

Table 6: Linearity statistics across multimodal datasets for Stable Diffusion 1.5.

The results show that for multimodal datasets (containing many classes like ImageNet or UCF-101), there is no evidence of a decrease in linearity compared to simple datasets (CelebA-HQ). For the same backbone model trained on a large multimodal dataset (LAION-5B), the slightly lower linearity on CelebA-HQ may be due to data processing techniques such as upscaling, which introduces blurriness and removes latent noise patterns from the Gaussian sphere.

When testing the linearity on OOD datasets for models trained on simple datasets, we observe a significant linearity drop. We compare models trained on multimodal data and tested on simple datasets. Results are computed using the same testing method as before and as described in Appendix B. Here, "trained → tested" means that the model is trained on the training set of the "trained" dataset and tested on the validation set of "tested".

For fair evaluation, we use the same model architecture (DDPM++) with the same training loss for these two pixel-space diffusion models. We find that for non-foundation models, OOD linearity drops significantly. The multimodal backbone has a slightly lower linearity score than the simple data backbone due to complexity in its training data (it trains on LAION-5B). The probability distribution of multimodal images plays an important role in linearity.

| Dataset (trained → tested) | ImageNet → FFHQ | FFHQ → ImageNet | FFHQ → FFHQ |
|---|---|---|---|
| $R^2$ | 0.934 | 0.938 | 0.995 |
| Cosine Similarity | 0.902 | 0.905 | 0.958 |

Table 7: Cross-dataset linearity comparison between multimodal and simple dataset training/testing. Results are computed using the same testing method as before and as described in Appendix B. Here, "trained → tested" means that the model is trained on the training set of the "trained" dataset and tested on the validation set of "tested".

# C    Limitations and Clarifications

## C.1    DDIM inversion not conforming to standard Gaussian distribution.

**1. Investigation of Sample Quality Drop.**    To investigate whether there is a sample quality drop, we use our CelebA-HQ validation set for additional experiments. This dataset is originally of size $256 \times 256$, and we upscale it to $512 \times 512$, so it becomes slightly blurry and sometimes gives inversions not perfectly on the sphere. We partition the CelebA-HQ data into two sets:

- **Not STG noise:** encoded noise with mean deviation $> 0.03$ or std deviation $> 0.03$ from standard Gaussian distribution.
- **STG noise:** all remaining samples.

We compute performance metrics of these images on Stable Diffusion with rMSE target $0.07$. The results are summarized below:

| Set | PSNR ↑ | MUSIQ ↑ | CLIP-IQA ↑ | SD ↑ |
|---|---|---|---|---|
| Not STG noise | 30.86 | 49.43 | 0.734 | 0.053 |
| STG noise | 30.10 | 49.74 | 0.731 | 0.054 |

Table 8: Comparison of image quality for STG vs. non-STG noise at rMSE target 0.07 on CelebA-HQ.

We do not find significant image quality differences between these two sets. Indeed, after our CCS interpolation at the 0.07 rMSE target, the interpolated noise of the non-STG noise group all falls within 0.01 difference between zero mean and unit standard deviation. Empirically, at low rMSE levels, the noise may be non-standard Gaussian, but the samples remain close to the input image. At higher rMSE levels, the interpolated noise becomes more standard Gaussian, resulting in good image quality.

**2. Verification of Gaussianity with Interpolation Strength.**    To verify that the Gaussian distribution becomes more standard with stronger interpolation, we compute the average deviation of mean (from 0) and deviation of variance (from 1) for the CelebA-HQ experiment. We find that as the interpolation strength $C_0$ increases, the deviation quickly narrows:

| $C_0$ | Deviation in mean | Deviation in std |
|---|---|---|
| 0.0 | 0.025 | 0.023 |
| 0.2 | 0.013 | 0.016 |
| 0.3 | 0.011 | 0.009 |
| 0.4 | 0.010 | 0.008 |
| 0.5 | 0.009 | 0.006 |
| 0.6 | 0.007 | 0.005 |

Table 9: Deviation of mean and standard deviation across interpolation strength $C_0$ in CelebA-HQ experiments.

**3. Discussion.** In our paper, $C_0$ is mostly between $0.3$ and $0.6$, so we do not worry much about the interpolated noise being non-standard Gaussian. We also conduct experiments at different rMSE target levels (with varying $C_0$) and observe that increasing interpolation strength may lead to slightly better image quality when the input image is not very good.

## C.2 Some other common confusions

There might be some misunderstanding of our method and experiments. To clarify:

- the baseline CCDF is just a special type of SDEdit.

- Our method **does not** specifically focus on improving quality in our main experiments in the main paper (the improvement in sample quality is a very nice add-on). Instead, it focuses on controllability (how to sample around a mean with a target MSE). So even though our performance gain is not that large, we achieve much better controllability over the edit strength which is measured as the distance between sample mean and the real input image (as demonstrated in Fig. 3).

# D    Additional Results and Experiments

In this section, we clarify some implementation details, providing more details on algorithms and visualization. We also provide more quantitative results and computational efficiency analysis.

## D.1    More Details on Controllable Sampling

**Experimental Set up** For pixel diffusion models, we use the first 50 images from the validation data from the FFHQ-256 [6] dataset. Then we set each image as the target mean and then sample 120 images (6000 images in total) with each target mean with a target rMSE (square root of average L-2 norm of the residuals between the sample and target mean) of 0.12. Then we test on the CIFAR-10 dataset. We randomly sample 20 images serving as target means, and then sample 120 images for each target mean with a target rMSE level of 0.11.

For Stable Diffusion, we use the SD1.5 checkpoint [31]. We study a more challenging scenario (degraded low-resolution input images with conditional text-guided latent diffusion model). We sample 50 images from the validation set from Celeba-HQ dataset with resolution $256 \times 256$, and then use bicubic upsampling to upscale it to $512 \times 512$. Note that SD1.5 is not trained on the Celeba-HQ dataset so this demonstrates the generalization capability of algorithms. We use the same prompt and CFG level in the linearity control experiments.

**Implementation Detail**
We follow Alg. 1 in implementing our methods for pixel diffusion models, and Alg. 3 for latent diffusion models. We take the pretrained models for FFHQ and CIFAR-10 from the improved/guided diffusion repos [30, 13] for the pixel diffusion experiments, and the Stable Diffusion 1.5 [31] for latent diffusion experiments. For LDMs, we set $t_0 = 45$, where $T = 50$ due to DDIM inversion performing worse with classifier-free guidance than unconditional models. We set the rMSE target to be 0.12, 0.11 for FFHQ and CIFAR-10 respectively, and 0.07 for Stable Diffusion experiments to test diverse control targets. The tolerance is set to be 0.01 in all cases. More details in the Appendix.
**Baselines** Since we are doing a novel task, we self-design the baselines with our proposed controller algorithm as an add-on.

- Gaussian Perturbation with Controller (GP-C): We add a Gaussian perturbation to the initial noisy image $x_{t0}$, where the perturbation scale is determined by our controller. This method resembles works that perform local editing [2].
- (Latent) Diffusion Posterior Sampling [6, 34] with controller (DPS-C): We perform posterior sampling with $x_0$ as the measurement. The scale of the gradient term in (L)DPS can control the randomness, so we design a controller based on this. Details in the Appendix.
- ILVR with controller (ILVR-C): the ILVR algorithm [4] is for sampling high quality images based on a reference image. The larger the downsampling parameter gives a better diversity, we dynamically adjust that parameter as by our controller algorithm. Since it is designed only for DDPM, we do not experiment it with LDMs. Details in the Appendix.

- Come-closer-diffuse-faster with controller (CCDF-C): CCDF use DDPM forward to find a starting noise at $t_0$, and then perform reverse sampling based on that noise [8]. We adjust $t_0$ based on our controller algorithm.
- Linear Interpolation with controller (LP-C): Replacing CCS spherical interpolation with linear interpolation

**Evaluation Metrics**

We first compute pixel-wise metrics to validate our hypothesis that sample mean is close to the target mean.

- PSNR (Peak Signal-to-Noise Ratio): quantifies the pixel-wise difference between the target mean and the sample mean.
- SD: the average of standard deviations of pixel intensities for each sampled image, which is used to measure the diversity of images.

Then we compute perceptual and reference-free metrics to measure the sample quality:

- MUSIQ [23]: measures the perceptual image quality, which focuses on low-level perceptual quality and is sensitive to blurs/noise/other distortions
- CLIP-IQA [40]: measures the semantic image quality, which is more higher-level than MUSIQ
- Inception Score (IS) [32]: is used in the CIFAR-10 dataset to further measure image quality and diversity. Since CIFAR-10 has a low resolution and images are blurry, we report IS score instead of MUSIQ and CLIP-IQA for CIFAR-10.

Additionally, we compute LPIPS between sampled image and target mean, this reflects how the samples are preserving source information even though the MSE of those samples are controlled for a fair comparison.

**Results**

We observe that our method achieves preserving more source information while generate superior quality images with sufficient diversity. The Table below shows our superior performance in this direction:

Table 10: Performance with Stable Diffusion with MSE level 0.07

| Methods | PSNR ↑ | LPIPS ↓ |
|---|---|---|
| GP-C | 23.02 | 0.306 |
| LDPS-C | 24.56 | 0.351 |
| CCDF-C | 27.66 | 0.318 |
| LP-C | 29.59 | 0.322 |
| **CCS (Ours)-C** | **30.29** | **0.252** |

Table 11: Performance with Pixel Diffusion on FFHQ with MSE level 0.12

| Methods | PSNR ↑ | LPIPS ↓ |
|---|---|---|
| GP-C | 18.88 | 0.596 |
| ILVR-C | 20.04 | 0.443 |
| DPS-C | 21.02 | 0.459 |
| CCDF-C | 23.52 | 0.461 |
| LP-C | 23.41 | 0.489 |
| **CCS (Ours)-C** | **25.13** | **0.332** |

Table 12: Performance with Pixel Diffusion on CIFAR-10 with MSE level 0.11

| Methods | PSNR ↑ | LPIPS ↓ |
|---|---|---|
| GP-C | 24.66 | 0.409 |
| DPS-C | 23.13 | 0.567 |
| CCDF-C | 24.63 | 0.529 |
| **CCS (Ours)-C** | **26.05** | **0.328** |

## D.2 Sampling Efficiency

We provide sampling speed and sampling NFE results in the table below. The sampling procedure consists of two steps:

1. **Controller tuning for statistical constraint.** This only needs to be performed once for each album. For our method and GP, we have an additional one-time single-image inversion step (which turns out to be very fast).

2. **Sampling with the tuned parameters.** Thanks to the linearity property, the binary search algorithm can efficiently find the feasible scale of perturbation, achieving the best controller tuning efficiency. Otherwise, the feasible scales may lie in a narrower region due to abrupt output changes or variability among samples, requiring more search rounds. We will include a detailed discussion of sampling efficiency in our revision.

We provide inference time for sampling 120 images around a target mean (with a tuning batch of size 20) for different methods below, using Stable Diffusion tested on one A40 GPU. The table reports the controller tuning NFE per batch, and sampling NFE per batch for each baseline.

| Method | Sample time / image | Controller Tuning NFE | Sampling NFE |
|---|---|---|---|
| GP-C | 1.66s | 96 | 45 |
| CCDF-C | 1.73s | 163 | **42** |
| LDPS-C | 5.84s | 234 | 50 |
| **CCS-C (Ours)** | **1.65s** | **94** | 45 |

Table 13: Sampling efficiency comparison on Stable Diffusion tested on one A40 GPU.

The advantage of CCDF-C (or in other words, SDEdit-C) is that it does not require DDIM inversion, and requires fewer timesteps for denoising. However, it needs more controller tuning rounds since the outputs can be highly sensitive to some timesteps.

## D.3 More Results on Adjusting rMSE Control Levels

We perform additional experiments with sampling quality benchmarks using rMSE targets from [0.05, 0.06, 0.07, 0.08, 0.09, 0.10] for Stable Diffusion on CelebA-HQ. We observe that our method consistently performs quite well (PSNR is for sample mean vs. target image, SD is for diversity). Numbers are reported in the order of CCS-C / CCDF-C (the stronger/best baseline).

| Target rMSE | PSNR ↑ | MUSIQ ↑ | CLIP-IQA ↑ | SD ↑ |
|---|---|---|---|---|
| 0.05 | 32.22/30.86 | 49.60/49.53 | 0.729/0.730 | 0.036/0.034 |
| 0.06 | 31.44/29.03 | 49.58/49.02 | 0.731/0.731 | 0.043/0.040 |
| 0.07 | 30.29/27.66 | 49.66/48.91 | 0.732/0.735 | 0.053/0.051 |
| 0.08 | 30.10/26.80 | 49.85/48.23 | 0.742/0.729 | 0.056/0.052 |
| 0.09 | 29.74/25.98 | 49.82/48.01 | 0.740/0.732 | 0.061/0.054 |
| 0.10 | 29.31/25.20 | 49.80/46.74 | 0.731/0.727 | 0.063/0.057 |

Table 14: Sampling quality comparison for different rMSE targets on CelebA-HQ of CCS-C v.s. CCDF-C: CCS on the left, CCDF on the right

We also perform experiments on the FFHQ dataset for rMSE levels of [0.09, 0.12, 0.15], summarized below:

| Target rMSE | PSNR ↑ | MUSIQ ↑ | CLIP-IQA ↑ | SD ↑ |
|---|---|---|---|---|
| 0.09 | 28.25/27.08 | 66.53/66.10 | 0.749/0.750 | 0.078/0.069 |
| 0.12 | 25.13/23.52 | 66.79/66.15 | 0.750/0.746 | 0.104/0.088 |
| 0.15 | 23.45/20.64 | 66.71/65.24 | 0.743/0.740 | 0.131/0.104 |

Table 15: Sampling quality results on FFHQ for different rMSE levels FOR CCS-C v.s. CCDF-C: CCS on the left, CCDF on the right

**Observations**

We observe several interesting phenomena in these additional experiments:

1. With more perturbation of noise (i.e., sample images farther from the target image), there is no decrease in sampling quality. Instead, there is a slight **increase in MUSIQ** for stable diffusion, meaning the image quality increases and looks sharper. One explanation is that the target image may not have good image quality or that the DDIM inversion is slightly apart from the Gaussian sphere. As the perturbation level increases, more Gaussian noise is interpolated, making the interpolated noise more "Gaussian" and improving image quality. Since Stable Diffusion is conditional and the input image is not from its training distribution, the DDIM inversion lies slightly off the Gaussian sphere. For FFHQ, however, the inversion is quite Gaussian, so there is no significant change in sample quality.

2. The sampled mean remains close to the target image as the rMSE level increases. We do not observe a sudden drop in controllability or diversity (PSNR and SD). However, with increasing rMSE targets, it becomes harder to control the sample mean close to the target mean, as demonstrated by declining PSNR. Nevertheless, this trade-off brings diversity improvement.

**D.4    More Details on improving sample quality.**

Based on the observation that interpolates with a Gaussian make a non-Gaussian random variable more Gaussian, previous work points out that this gives a higher likelihood [30]. Hence, we propose to apply P-CCS on every step of reverse sampling with a very small interpolation factor. The algorithm is stated at Alg. 5. We observe significant gain in sample quality when testing on T2IBench [21]. Table. 16 shows the quantitative performance.

Table 16: Image Quality Scores for with and without P-CCS purified on T2IBench

| Metric | With | Without |
|---|---|---|
| MUSIQ Score | **55.499** | 52.951 |
| CLIP-IQA Score | **0.541** | 0.530 |

**D.5    More Algorithms**

We describe more details for the proposed P-CCS (Partial inversion CCS sampling) algorithms for different applications including constrained sampling, precise image editing and improving sampling quality.

Alg. 3 demonstrates using P-CCS for constrained sampling based on Stable Diffusion.

Alg. 4 demonstrates using P-CCS for precise image editing based on Stable Diffusion.

Alg. 5 demonstrates using P-CCS for improving the sample quality, instead of controllability.

---

**Algorithm 3** P-CCS for Constrained Sampling
---
**Requires:** target mean $x_0$, perturbation scale $C_0$, inversion time steps $t_0$, Encoder $\mathcal{E}$ and Decoder $\mathcal{D}$, a prompt $c$.

**Step 0:** Compute $\mathbf{z}_0 = \mathcal{E}(x_0)$, then compute the DDIM inversion of $\mathbf{z}_0$, i.e. $\mathbf{z}_T = \text{DDIM}^{-1}(\mathbf{z}_0, 0, t_0, c)$

**Step 1:** Compute the noise from $\mathbf{z}_{t_0}$, by $\epsilon_{t_0} = \mathbf{z}_{t_0} - \sqrt{\alpha_{t_0}} \cdot \mathbf{z}_0$

**Step 2:** Sample noise $\epsilon \sim \mathbb{N}(0, 1 - \alpha_t)$. Then compute

$$\theta = \cos^{-1}\left( \frac{\epsilon \cdot \epsilon_{target}}{||\epsilon||_2 ||\epsilon_{target}||_2} \right)$$

**Step 3:**
Compute $\epsilon'_{t_0}$ using spherical interpolation formula:

$$\epsilon'_{t_0} = \frac{\sin(C_0)}{\sin(\theta)} \cdot \epsilon + \frac{\sin(\theta - C_0)}{\sin(\theta)} \cdot \epsilon_{t_0}$$

**Step 4:** Compute $\mathbf{z}'_{t_0} = \sqrt{\alpha_{t_0}} \cdot \mathbf{z}_0 + \epsilon'_{t_0}$

**Step 5:** Output sample $\mathbf{x}'_0 = \mathcal{D}(\mathbf{z}'_0) = D(\text{DDIM}(\mathbf{z}'_{t_0}, t_0, 0, c))$

---

---

**Algorithm 4** P-CCS for Precise Image Editing
---
**Requires:** target mean $x_0$, perturbation scale $C_0$, inversion time steps $t_0$, Encoder $\mathcal{E}$ and Decoder $\mathcal{D}$, source prompt $c_s$, target prompt $c_t$.

**Step 0:** Compute $\mathbf{z}_0 = \mathcal{E}(x_0)$, then compute the DDIM inversion of $\mathbf{z}_0$ with the source prompt, i.e. $\mathbf{z}_{t_0,s} = \text{DDIM}^{-1}(\mathbf{z}_0, 0, t_0, c_s)$, and the DDIM inversion with the target prompt, i.e. $\mathbf{z}_{t_0,t} = \text{DDIM}^{-1}(\mathbf{z}_0, 0, t_0, c_t)$

**Step 1:** Compute the noise from $\mathbf{z}_{t_0,s}$, by $\epsilon_{t_0,s} = \mathbf{z}_{t_0,s} - \sqrt{\alpha_{t_0}} \cdot \mathbf{z}_0$

**Step 2:** Compute the noise from $\mathbf{z}_{t_0,t}$, by $\epsilon_{t_0,t} = \mathbf{z}_{t_0,t} - \sqrt{\alpha_{t_0}} \cdot \mathbf{z}_0$

**Step 3:** Compute

$$\theta = \cos^{-1}\left( \frac{\epsilon_{t_0,s} \cdot \epsilon_{t_0,t}}{||\epsilon_{t_0,s}||_2 ||\epsilon_{t_0,t}||_2} \right)$$

**Step 3:**
Compute $\epsilon'_{t_0}$ using spherical interpolation formula:

$$\epsilon'_{t_0} = \frac{\sin(C_0)}{\sin(\theta)} \cdot \epsilon_{t_0,s} + \frac{\sin(\theta - C_0)}{\sin(\theta)} \cdot \epsilon_{t_0,c}$$

**Step 4:** Compute $\mathbf{z}'_{t_0} = \sqrt{\alpha_{t_0}} \cdot \mathbf{z}_0 + \epsilon'_{t_0}$

**Step 5:** Output sample $\mathbf{x}'_0 = \mathcal{D}(\mathbf{z}'_0) = D(\text{DDIM}(\mathbf{z}'_{t_0}, t_0, 0, c_t))$, which is the precisely edited image with strength given by $C_0$.

---

### D.6 More Figures

Fig. 6 demonstrates the generated personalized album for CCS. Fig. 7 demonstrates example of applying P-CCS with SD1.5 on the Celeba-HQ dataset, we demonstrate that our algorithm can work well on in-the-wild images which are very different from the training.

Fig. 10 demonstrates the linear trend for each target mean on the FFHQ dataset.

Fig. 8,9 demonstrates an example of image editing controlled sampling with Alg. 4.

Fig. 11 demonstrates the linearity drop with OOD data.

---

**Algorithm 5** P-CCS for Improving Sampling Quality

---

**Requires:** target mean $x_0$, perturbation scale $C_0$, inversion time steps $t_0$, Encoder $\mathcal{E}$ and Decoder $\mathcal{D}$, condition $c$.

**Step 0:** Start by computing $\mathbf{z}_0 = \mathcal{E}(x_0)$, then compute the DDIM inversion of $\mathbf{z}_0$ with the condition, i.e. $\mathbf{z}_{t_0} = \text{DDIM}^{-1}(\mathbf{z}_0, 0, t_0, c)$,

**While $t_0 > 0$: Step 1:** Compute the noise from $\mathbf{z}_{t_0}$, by $\epsilon_{t_0} = \mathbf{z}_{t_0} - \sqrt{\alpha_{t_0}} \cdot \hat{\mathbf{z}}_0(z_T)$ using Tweedie's formula to compute $\hat{\mathbf{z}}_0(z_T)$.

**Step 2:** Sample a noise $\epsilon \in N(0, 1 - \alpha_t)$.

**Step 3:** Compute

$$\theta = \cos^{-1}\left(\frac{\epsilon_{t_0} \cdot \epsilon}{||\epsilon_{t_0}||_2 ||\epsilon||_2}\right)$$

**Step 3:**

Compute $\epsilon'_{t_0}$ using spherical interpolation formula:

$$\epsilon'_{t_0} = \frac{\sin(C_0)}{\sin(\theta)} \cdot \epsilon_{t_0} + \frac{\sin(\theta - C_0)}{\sin(\theta)} \cdot \epsilon$$

**Step 4:** Compute $\mathbf{z}'_{t_0} = \sqrt{\alpha_{t_0}} \cdot \mathbf{z}_0 + \epsilon'_{t_0}$

**Step 5:** Reverse Sampling using DDIM formula, and the modified $\mathbf{z}_{t_0}$, $t_0 = t_0 - 1$

**End While**

**Step 6:** Output sample $\mathbf{x}'_0 = \mathcal{D}(\mathbf{z}'_0) = D(\text{DDIM}(\mathbf{z}'_{t_0}, t_0, 0, c_t))$, which is the purified (improved) image with purification strength given by $C_0$.

---

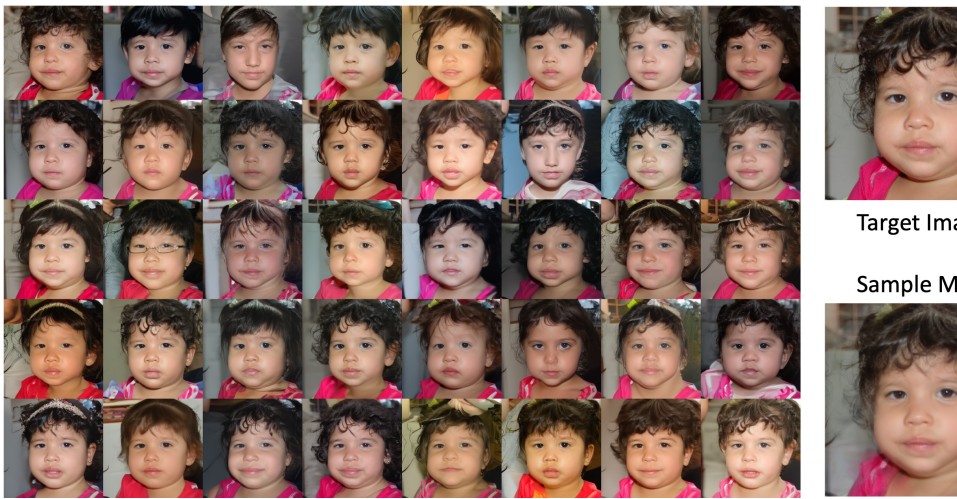

Figure 6: a demo of sampled album with CCS algorithm on FFHQ dataset. Note that the sample mean is almost the same as the input image.

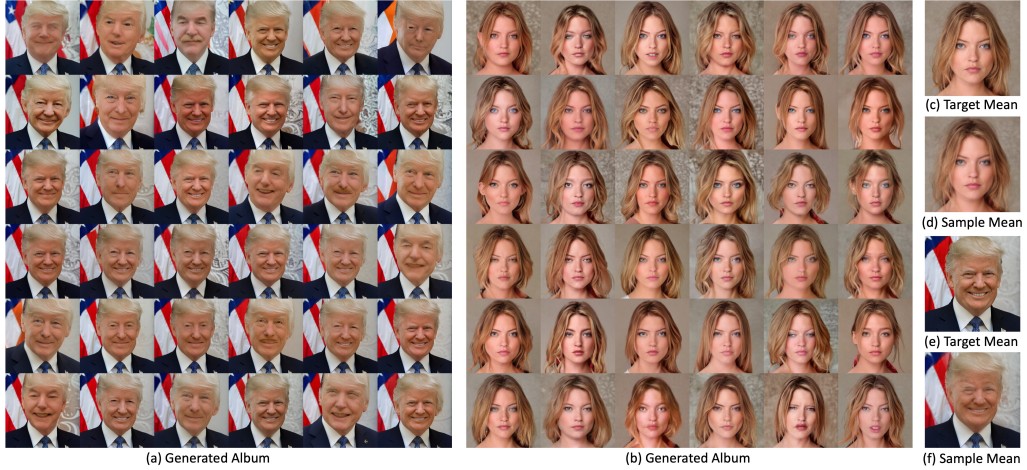

Figure 7: CCS-CT Sampled Images with Stable Diffusion 1.5. (a) Samples with an in-the-wild target mean (e) and a target rMSE 0.09; (b) Samples with a target mean (d) from Celeba-HQ dataset and a target rMSE 0.07; (d): sample mean from (a); (f): sample mean from (b).

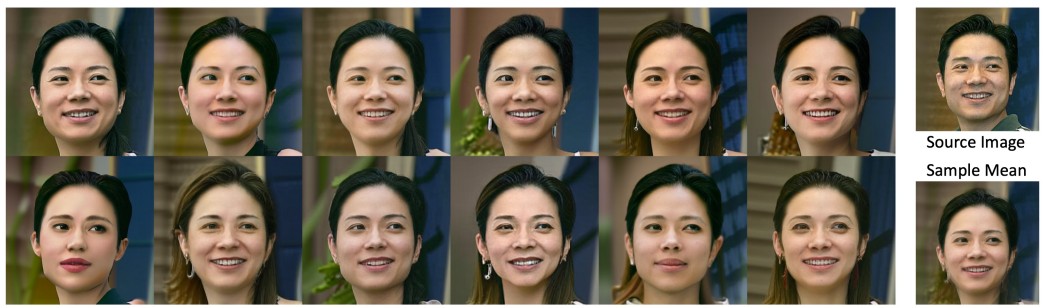

Figure 8: Image Editing Samples with Stable Diffusion 1.5, the source prompt is given by 'a high-quality portrait of a man', and the target prompt is given by 'a high-quality portrait of a woman', the target MSE level is given by 0.10

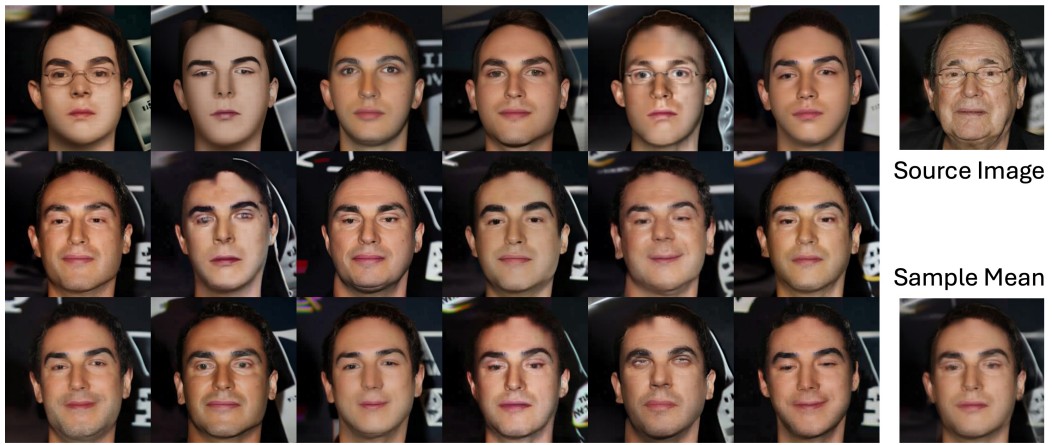

Figure 9: Image Editing Samples with Stable Diffusion 1.5, the source prompt is given by 'a high-quality portrait of an old man', and the target prompt is given by 'a high-quality portrait of a young man', the target MSE level is given by 0.09

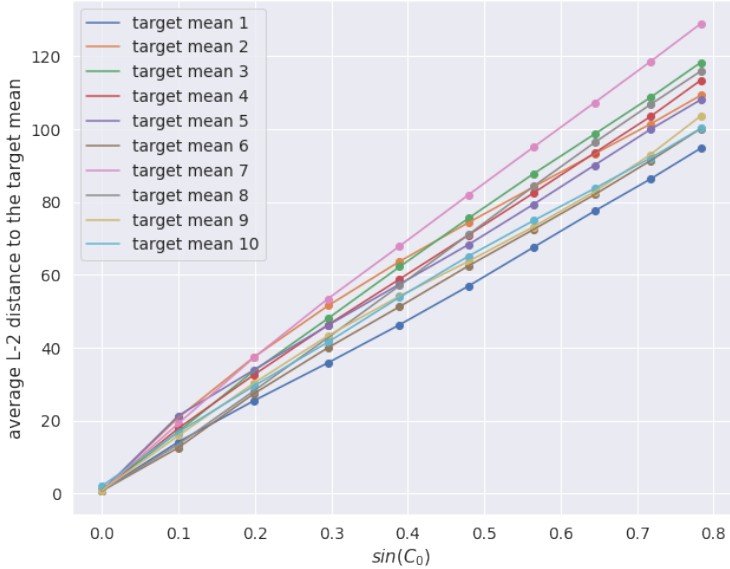

Figure 10: We first sample 50 images around each target mean in the FFHQ dataset. We then obtain the DDIM inverse of each target mean, and then add spherical perturbation to it. When the scale of perturbations $\sin(C_0)$ increases, the average of norms of the residuals between each sample and the target mean approximately increases linearly.

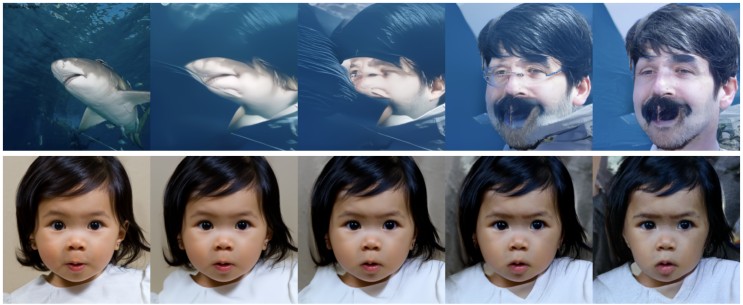

Figure 11: We demonstrate that there is sudden change in output when testing on OOD input perturbed with increasing Gaussian noise. The backbone model is a pixel diffusion model (DDPM++), which is only trained on FFHQ. The top row uses an OOD input from ImageNet, and the bottom row uses an image from FFHQ validation set input. $C_0$ from left to right: 0.0, 0.2, 0.4, 0.6, 0.8

