# OpenReview forum: "CCS: Controllable and Constrained Sampling with Diffusion Models via Initial Noise Perturbation"
_NeurIPS.cc/2025/Conference — NeurIPS 2025 poster_

### Official Review · Reviewer_qRjk · 2025-06-30

**Clarity:** 3
**Significance:** 2
**Originality:** 2
**Rating:** 5
**Confidence:** 3

**Summary:**

- Introduces new problem of ensuring that generated samples from diffusion model are within specified range of target mean
- Analyze and characterize linearity of DDIM sampling process with respect to small perturbations in initial noise, both theoretically and empirically
- Propose an algorithm based on spherical interpolation to ensure generated samples from DDIM fall within certain MSE range of target mean, with hyper-parameter to adjust for diversity in generated samples
- Compare against self-implemented baselines to demonstrate effectiveness of proposed algorithm

**Questions:**

*Empirical*
1. In the experiments, it seems that the target rMSE from the given mean is set to be .12 for Table 2, and .07 for Table 3. Why are these selected as target rMSE? It seems clear that the proposed method is better than baselines at attaining this target rMSE, but it is not explained why attaining this specific value is important or relevant. Does it have to do with balancing fidelity to the target and reasonable diversity? It would be good to explain why targeting this specific value is reasonable in the context of this task.

2. As explained, Algorithm 2 performs a binary search to achieve a target level of diversity by repeatedly calling Algorithm 1, which is where the spherical interpolation is applied to the DDIM sampling process. But this Controller Tuning algorithm seems to be expensive, as it requires multiple calls of the DDIM sampling process. I am not familiar with the baselines, so I am uncertain as to whether this is a reasonable expense. It would be very helpful to provide something along the lines of an average NFE per generated album for the methods to compare the running cost — while CCS outperforms the baselines, this outperformance may not be as significant if it turns out that it is significantly more expensive than the other methods.

*Significance*
1. Why is it important to be able to control the mean / distance from the mean? I am not fully convinced of the significance of this problem.

Currently, I am leaning towards rejection. If the significance is clarified, along with some of the technical details, I am happy to recommend acceptance.

**Ethical Concerns:**

["NO or VERY MINOR ethics concerns only"]

**Final Justification:**

I decided to increase my score by two points from a 3 to a 5. I believe that the strengths of the paper are as follows:

1. Well-motivated problem within diffusion and image generation

2. Theoretical explanation for linearity effect and DDIM inversion

3. Practical effectiveness in both level of control and computational cost.

Prior to their rebuttal, I was only confident in the second point. However, the authors have convinced me of the first and third point. The explanation of the problem's relevance in their rebuttal convinced me of the first point; and the pSNR ablation + NFE comparison convinced me of the third point. I was willing to increase my score if I was convinced of either point, but as I have been convinced of both points, I felt that this paper warranted higher than a 4. I feel that this paper would be a good contribution to NeurIPS.

**Limitations:**

The authors acknowledge that their method only controls the target mean and spread from the target mean. However, the abstract and the introduction make it seem as if they are capable of controlling various statistical properties. I think it may be better to align the introduction and the abstract with what is actually shown, as the current method can only control the mean / distance from the mean and not higher level moments.

**Paper Formatting Concerns:**

No paper formatting concerns were noted, but I could have missed some issues.

**Quality:**

3

**Strengths And Weaknesses:**

*Empirical*

The empirical comparisons seem to be thorough — the baselines selected seem relevant towards the specific task. The experimental setup itself seems well motivated. The appendix contains sufficient details to reproduce the experiments, or at least understand all of the specific setup parameters.

*Theoretical*

The theoretical contributions seem quite clear, and the proofs in the appendix are easy to follow. I liked the core finding that the overall change is at most linear, given that small changes are approximately linear in effect.

*Algorithmic*

The algorithm boxes themselves are laid out nicely and are easy to follow. However, I think it would be good to discuss the computational cost of the proposed algorithm in comparison to the baselines, as otherwise it is difficult to make fair comparisons.

---

> ### Author Rebuttal · Authors · 2025-07-31
>
> We thank the reviewer for the careful and insightful review!
>
>
> ***1. Q: Why is it important to be able to control the mean / distance from the mean?***
>
> > Thanks the reviewer for raising this important question. This question points to the core novelty and impact of our work.
>
> > We may be confused about why we want to control the distance of the samples to the target image? or why we want the distance between sample mean and target image to low? Intuitively a sample mean should be blurry since the averaging of different image features should be blurry. Then why computing PSNR between sample mean and target image make sense?
>
> > 1. For image editing, there are numerous work built on DDIM inversion. However, to our best knowledge, there is no systematic quantitative study and benchmark on why DDIM inversion is better than other techniques such as SDEdit [1] or Classifier-guidance (DPS) [2]. Most of the papers give vague intuitions such as: "Other techniques struggle to accurately preserve the input image details." (comparing with SDEdit) [3].  A fundamental issue here is how much percent the input image details are lost by techniques such as SDEdit.
>
> > One may think that we can compute the similarity (such as LPIPS) between sampled/edited images and the input image to measure how much the input image features are preserved. However, this metric can be misleading since if the samples and the input images are identical, we can have 100 percent similarity. So here comes our ***first novelty***: we control the average distance between the samples and the input image (rMSE) to be fixed and then compute the sample diversity and feature preservation.
>
> > 2. Then, to measure the source feature preservation capability with fixed rMSE level, we may want to know what specific features can be preserved? Also are there some ***common*** features been preserved for each sample consistently? These questions are important for image editing users since they may want human faces been preserved but not the background or other textures. If we compute the average LPIPS distance between samples and the input image, there is no way we can learn about the common features preservation. Then here comes our ***second novelty***: measuring the distance between sample mean and input image.
>
> >Thank to the linearity property revealed in our paper, we found that the sample mean using CCS sampling is surprisingly sharp and not blurry. We then have an idea that the sample mean should contain information about the common features in all samples, so if we directly compute the distance between sample mean and input image, then we can reveal both how well samples contain details from input image and how well the common features being preserved in all samples. A visual example can be found in Fig.6 and Fig.7 in the Appendix (for example in Fig.6 all sample images have eyes looking similar to the input image, which is revealed by the sample mean (otherwise the sample mean will be blurry)).
>
> >3. Furthermore, users may want fine-grained control over source feature preservation. Suppose that we are building a photo editing/creation app, and users want to generate new photos based on their input photo. we may need a sliding bar functionality that when a user slides that bar, the more different generated photos are from the original photo. Different users may have different preferences about how the generated photos look similar to their original photos in texture, facial expression, hair style and so on. Providing this sliding bar functionality will be very useful to satisfy their needs.
>
> >This is the motivation about why we propose this novel problem. We believe this problem and evaluation method can be impactful for image editing or conditional image generation. We will revise our paper to explain and highlight this.
>
>
>
>
>
> ***2. Q: Why are these selected as target rMSE? Does it have to do with balancing fidelity to the target and reasonable diversity?***
>
> > Thanks the reviewer for raising this important and interesting question.
> The target rMSE in this paper is set for best visualization quality (let readers see samples close to source image and have sufficient variations). Investigating how performance changes with different rMSE is interesting and we will add in revision.
>
> > We perform additional experiments with sampling quality benchmark with rMSE targets from [0.05, 0.06, 0.07,0.08,0.09, 0.10] for Stable Diffusion on CelebA-HQ. We observe that our method consistently performs quite well (PSNR is for sample mean v.s. target image, SD is for diversity). (numbers are reported in the order of CCS-C/CCDF-C (the strongest/best baseline))
>
> | Target rMSE | PSNR↑  | MUSIQ↑ | CLIP-IQA↑| SD ↑|
> | ---------| ---- | ----- | -------| -----|
> |0.05 | ***32.22***/30.86   | ***49.60***/49.53 | 0.729/***0.730*** | ***0.036***/0.034|
> | 0.06 | ***31.44***/29.03| ***49.58***/49.02 | ***0.731***/0.731| ***0.043***/0.040|
> | 0.07 | ***30.29***/27.66 | ***49.66***/48.91 | 0.732/***0.735***| ***0.053***/0.051|
> | 0.08 | ***30.10***/26.80| ***49.85***/48.23 | ***0.742***/0.729| ***0.056***/0.052|
> | 0.09 | ***29.74***/25.98 | ***49.82***/48.01 | ***0.740***/0.732| ***0.061***/0.054|
> | 0.10 | ***29.31***/25.20 | ***49.80***/46.74 | ***0.731***/0.727| ***0.063***/0.057|
>
>
> > We also perform experiments on FFHQ dataset for rMSE levels of [0.09, 0.12, 0.15], and summarize the results below:
>
> | Target rMSE | PSNR  | MUSIQ | CLIP-IQA| SD |
> | ---------| ---- | ----- | -------| -----|
> |0.09 | ***28.25***/ 27.08   | ***66.53***/66.10 | 0.749/***0.750*** | ***0.078***/0.069|
> | 0.12 | ***25.13***/23.52| ***66.79***/66.15 | ***0.750***/0.746 | ***0.104***/0.088|
> | 0.15 | ***23.45***/20.64 | ***66.71***/65.24 | ***0.743***/0.740| ***0.131***/0.104|
>
>
> > We observe several intersting phenomenon in this additional experiment:
> > 1. With more perturbation of noise (the sample images farther from the the target image), there is no decrease in sampling quality. Instead, there is slight increase in MUSIQ for stable diffusion which means that image quality increases and looks sharper. One explanation for this is that the target image may not have good image quality or the DDIM inversion is slightly apart from the sphere, so when increasing the perturbation level, more Gaussian noise is interpolated so the interpolated noise becomes more "Gaussian" and the image quality may improve. Since the Stable Diffusion is a conditional model and the input image is not from its training data distribution, we observe that the DDIM inverted is slightly off the Gaussian sphere. But for FFHQ, the inverted is quite Gaussian, so there is no significant change in sample quality.
> > 2. the sampled mean is still close to the target image when increasing rMSE level. We also do not observe sudden drop in controllability and diversity (PSNR and SD). However, with increasing rMSE target, it becomes harder to control the sample mean close to target mean, as demonstrated by declining PSNR. Nevertheless, this trades off with diversity improvement.
>
>
> ***3. Q: But this Controller Tuning algorithm seems to be expensive, as it requires multiple calls of the DDIM sampling process.. It would be very helpful to provide something along the lines of an average NFE per generated album for the methods to compare the running cost***
>
> > Thanks reviewer for raising this! This is also an important question. We will add this in revision. Our method actually achieves the best overall efficiency. The controller tuning can be costly fror our goal, but the zero-order controller should still be one of the best choices compared to first-order gradient searches.  We further use binary search to improve time complexity to log scale, which should be faster than other zero-order methods [4].
>
> > We provide sampling speed and sampling NFE in the table below. The sampling procedure consists of two steps:
> > 1. controller tuning for statistical constraint. This only needs to perform once for each album. For our method and GP, we have an addition one-time single-image inversion step (turns out to be very fast).
> > 2. sampling with the tuned parameters.
> > Thanks to the linearity property, the binary search algorithm can find the feasible scale of perturbation efficiently, and we achieve the best controller tuning efficiency. Otherwise, the feasible scales may lie in a much narrower region due to abrupt change in output or vary a lot for different samples, and we may need more rounds searches to find them. We will add a discussion about sampling efficiency in our revision. We provide the inference time for sampling 120 images around a target mean for different methods below for Stable Diffusion tested on one A40 GPU. We also provide the inversion NFE, controller tuning NFE per image, and sampling NFE per image for different baselines.
>
> |Method	|sample time per image| total inversion NFE | total controller tuning NFE | sampling NFE |
> | -------| -----| ------| ------|-----|
> |GP-C|	1.66s|  45 | 96 | 45 |
> |CCDF-C|	1.73s|  ***0*** | 163 | ***42*** |
> |LDPS-C|	5.84s|  ***0*** | 234 | 50 |
> |CCS-C (Ours)|***1.65s***| 45 | ***94*** | 45 |
>
> The advantage of CCDF-C or in other words (SDEdit-C) is that it does not require DDIM inversion, and requires fewer timesteps for denoising, but it needs more controller tuning rounds since the outputs can be very sensitive to some timesteps.
>
> ****4. Q: Limitations: introduction not clear****
> > Thanks for pointing out! We will revise our introduction to "We study controllable sampling with mean close to a target mean at a target diversity level for diffusion models."
>
> [1] SDEdit: Guided Image Synthesis and Editing with Stochastic Differential Equations. ICLR 2022
>
> [2] Diffusion Models Beat GANs on Image Synthesis. NeurIPS 2021
>
> [3] Null-text Inversion for Editing Real Images using Guided Diffusion Models. CVPR 2023
>
> [4] Inference-Time Scaling for Diffusion Models beyond Scaling Denoising Steps. CVPR 2025

---

> ### Author Response · Authors · 2025-08-04
> **Revision of our manuscript based on reviewer's suggestions**
>
> Dear Reviewer qRjk,
>
> Thanks for your valuable feedback. We have revised our manuscript based on your suggestions. We highlight some of our core changes here:
>
> 1. We revise the sentence in abstract Line 9-12 from
> ```we propose a novel Controllable and Constrained Sampling method (CCS) together with a new controller algorithm for diffusion models to sample with desired statistical properties while preserving good sample quality.```
> to:
> “we propose a novel Controllable and Constrained Sampling (CCS) method, along with a new controller algorithm for diffusion models, that enables control over both the proximity of individual samples to a target image and the alignment of the sample mean with the target, while preserving good sample quality”
>
> 2. We revise the introduction Line 42-44 to highlight the purpose of our work and our contribution from:
> ```Based on the spherical interpolation to perturb the initial noise vector, we propose a novel controllable and constrained sampling method (CCS) together with a new controller algorithm for diffusion models to sample with desired statistical properties while preserving high quality and adjustable diversity.```
>
> to
>
> “Based on the spherical interpolation to perturb the initial noise vector, we propose a novel controllable and constrained sampling method (CCS) for diffusion models to sample with a target rMSE level, enabling the sample mean to be close to the target image, while preserving high quality and adjustable diversity.
> The motivation for this task stems from a fundamental need in image editing and controllable generation: preserving key source features while allowing controlled variation. However, few works benchmark the sample quality and key feature preservation at a target controlled variation level. Our first key idea is to fix the average distance (rMSE) between samples and the target, enabling a fair comparison of sample diversity and feature preservation. Our second insight is to evaluate the distance between the sample mean and the target image, which reveals how well common features are preserved. In addition, our CCS algorithm enables a user-controllable “diversity slider”: a tool that adjusts how far generated samples deviate from the input image. This fine-grained control over similarity can be vital for practical applications like photo editing apps.”
>
> 3. We revise our Conclusion in Section 6, line 344 to include the limitation: "The limitation of our work includes 1. controlling statistical properties other mean and MSE is left as future work. 2. There might be some artifact samples that exhibit overlapping patterns. 3. DDIM inversion may not be perfectly standard Gaussian, which may hurt sample quality. Nevertheless, we demonstrate that our algorithm can remedy this issue."
>
> 4. In the experiment section (Section 5.2), we provide a rationale of choosing rMSE target levels: "we report the results for Stable Diffusion 1.5 on CelebA-HQ dataset with rMSE level 0.07, for pixel-based diffusion models on FFHQ dataset with rMSE level 0.12. We choose these levels because the generated albums at these levels demonstrate both sufficient variation and closeness to the original input image. We report additional results on the benchmark with different rMSE levels in the Appendix."
>
> 5. We add a section about the computational cost in the Appendix. In addition, for clarification, in our rebuttal,
> > 1. The total inversion NFE means the NFEs required to invert a single image
> > 2. The total controller tuning NFE means the average NFEs required to tune the controller for sampling with a target diversity level. Here we use a batch of 12 images for tuning. So it is #controller NFE x 12 for total number of diffusion model functional evaluations.
> > 3. Sampling NFE means the average NFEs to sample an image in the sampling stage. If sampling 120 images, the total number of of diffusion model functional evaluations is #sampling NFE x 120.
>
> > We try to clarify all details of evaluation in our revision.
>
> There are also some typos in our rebuttal (due to the tight rebuttal timeline).
>
> 1. The sentence "why we want the distance between sample mean and target image to low?" should be "why we want to control the distance between sample mean and the target image to be small".
>
> 2. The sentence "when a user slides that bar, the more different generated photos are from the original photo" should be "As the user moves the slider, the generated photos can become increasingly different from the original photo."

---

> > ### Comment · Reviewer_qRjk · 2025-08-05
> >
> > Thank you for your detailed response to my questions. I think the explanation towards the target rMSE makes sense, and the extra ablations helped illuminate some interesting trends. Additionally, I think that the subsequent revisions mentioned will make the paper much stronger, especially the clarification for the computational cost. Now that it is clear that the proposed algorithm is actually cheap compared to previous methods, the empirical performance seems much stronger.
> >
> > I also think that the motivation is much more clear -- the problem of generating diversity while maintaining fidelity to core features is a very well motivated problem to examine, and I think the CCS algorithm makes much more sense.
> >
> > For these reasons, I am happy to increase my score as all my primary concerns have been addressed.

---

> ### Author Response · Authors · 2025-08-06
>
> Dear Reviewer qRjk,
>
> Thank you for the encouraging feedback! We really appreciate your thoughtful question regarding the scope of our work, as it has helped us better articulate and refine our motivation. Your question on the limitation of our work also prompts us to think about more complicated settings (such as preserving more statistical features or more fine-grained controlling). We believe that our direction could be beneficial for future work in improving controllable sampling with large-scale diffusion models at the post-training stage or test time with our newly proposed task and evaluation method.

---

### Official Review · Reviewer_ivEh · 2025-07-02

**Clarity:** 3
**Significance:** 2
**Originality:** 3
**Rating:** 4
**Confidence:** 3

**Summary:**

This paper investigates the linear relationship between the initial noise and the generated outputs of diffusion models under DDIM sampling. The authors provide both theoretical and empirical evidence to support this linearity property. Based on this insight, they propose a method called CCS, which enables the generation of desired results under specific conditions such as image editing, personalized image generation, and quality improvement.

**Questions:**

An ablation study comparing the proposed spherical interpolation with simpler alternatives such as linear or additive interpolation could further validate the design choice.
It is unclear whether the initial noise obtained through DDIM inversion always conforms to a standard Gaussian distribution. If not, how sensitive is the proposed method to such deviations? An analysis of its robustness in such cases would be helpful.
I am also curious whether the method could be extended to enable fine-grained image editing—for example, preserving the identity of a character while modifying the surrounding environment. This seems related to selectively controlling noise components: keeping certain directions fixed while perturbing others more strongly.

**Ethical Concerns:**

["NO or VERY MINOR ethics concerns only"]

**Final Justification:**

The reviewer answered to my questions and I am keeping this positive score.

**Limitations:**

The authors claim they discuss limitations in the conclusion section, but I could not find it.

**Paper Formatting Concerns:**

good

**Quality:**

3

**Strengths And Weaknesses:**

Strengths:

    The observed linearity phenomenon is interesting and novel.
    The authors provide both theoretical analysis and empirical validation.
    Extensive experiments are conducted across various applications to demonstrate the effectiveness of the proposed method.

Weaknesses:

    The proposed CCS method relies on spherical interpolation. An ablation study comparing it with alternative interpolation strategies (e.g., linear or additive perturbations) would strengthen the justification for this design choice.
    The assumption that DDIM-inverted noise lies on a Gaussian hypersphere is not rigorously supported. In practice, the inversion result may deviate from a standard Gaussian distribution, potentially affecting the method’s reliability.
    The paper does not discuss limitations or provide visualizations of failure cases, which would help clarify the method’s robustness and boundary conditions.

---

> ### Author Rebuttal · Authors · 2025-07-31
>
> We thanks the reviewer for the careful and insightful review!
>
> ***1. Q: An ablation study comparing it with alternative interpolation strategies***
> > A: Thanks, this is also an important question we have. We perform additional experiments that compares the performance linear interpolation for the pixel diffusion experiments on FFHQ and Stable Diffusion experiments on CelebA-HQ following the same setting in the main paper (rMSE target = 0.07 for CelebA-HQ, and 0.12 for FFHQ, both using the same controller).  We will include the new results in our paper revision.
>
> >Results showing that through spherical interpolation we achieve better controllability demonstrated by higher PSNR (between sample mean and target image), better diversity, and better image quality. Visually, using linear interpolation causes very blurry images.
> We include the results for Stable Diffusion on CelebA-HQ here (PSNR for controllability, SD for diversity, MUSIQ for image visual quality, and CLIP-IQA for image semantic quality):
> | Method | PSNR↑ | MUSIQ↑ | CLIP-IQA↑ | SD↑ |
> | ------| -----| -----| -----| ----|
> | CCS-CT-C | 30.29 | 49.66 | 0.732 | 0.053|
> |Linear interpolation-C | 29.59 | 42.98 | 0.722 | 0.032|
>
> > Also the results on FFHQ:
> | Method | PSNR | MUSIQ | CLIP-IQA | SD |
> | ------| -----| -----| -----| -----|
> | CCS-CT-C | 25.13 | 66.79 | 0.750 | 0.104|
> |Linear interpolation-C | 23.41 | 53.26 | 0.731 | 0.057|
>
> > The result show severe degradation in MUSIQ and diversity, with significant drop in PSNR, suggesting linear interpolation leads to worse quality images with decreased diversity, and less controllability.
>
>
> > To further support this point, we provide additional theoretical analysis as below to justify why simple linear interpolation does not work well for sampling:
> Formally, let $||\Delta x||_2 = ||x||_2$, and its direction is uniformly distributed, and $\mathbb{E}[\Delta x] = 0$, for $0 < \alpha < 1$, we have $$\mathbb{E} [ || \alpha x + (1-\alpha) \Delta x ||_2^2] < || x ||_2^2$$
>
> > Proof: we have $||\alpha x + (1-\alpha) \Delta x ||_2^2 = \alpha^2 || x ||_2^2 + 2\alpha(1-\alpha)(x \cdot \Delta x) + (1-\alpha)^2 || \Delta x ||_2^2$, since we have $\mathbb{E}[\alpha(1-\alpha)(x \cdot \Delta x)] = \alpha(1-\alpha) x \cdot \mathbb{E}[\Delta x] = 0$, we have $\mathbb{E} [ || \alpha x + (1-\alpha) \Delta x ||_2^2] = \alpha^2 || x ||_2^2 + (1-\alpha)^2 || \Delta x ||_2^2 = (\alpha^2 + (1-\alpha)^2) || x ||_2^2$, now since $2\alpha (\alpha - 1) < 0$, we have $(\alpha^2 + (1-\alpha)^2) < 1$, and we can reach the conclusion that using simple linear interpolation cannot preserve the norm leading to falling apart from the Guassian sphere. On why not falling apart from Gaussian sphere may lead to degraded image quality. We argue that if initial noises not on spherical surface, this means that the noise level is either too low or too high by Proposition 4 in our paper, which both leads the trained denoiser of the diffusion model not effectively removing the noise effectively (noisy image) or removing too much noise (blurry image).
>
> ***2. Q: DDIM inversion not conforming to standard Gaussian distribution. Are there any sensitivity to those deviations***
> > Thanks reviewer for pointing this out. We agree that the DDIM inversion may not follow standard Gaussian distribution. However, through our spherical interpolation method, we are able to make the interpolated noise more "standard Gaussian". The risk is that spherical interpolation for a non-standard Gaussian and a standard Gaussian noise may still be quite non-standard Gaussian, and cause the image sample quality degradation. However, we observe that in practice this is rarely the case.
>
> > 1. To investigate whether there is sample quality drop, we use the our CelebA-HQ validation set in our paper for additional experiments. This dataset is originally of size 256x256, we upscale it to 512x512, so it is blurry and sometimes gives inversion not on the sphere. We partition the CelebA-HQ into two sets:  the encoded noise that has a mean deviates from 0 by more than 0.03 or std deviates from 1 by more than 0.03 (not STG noise), and the rest. We compute the performance metric of these images on Stable Diffusion with rMSE target 0.07 and summarized below:
> | Set | PSNR | MUSIQ | CLIP-IQA | SD |
> | ------| -----| -----| -----| -----|
> | not STG noise | 30.86 | 49.43 | 0.734 | 0.053|
> |STG noise | 30.10 | 49.74 | 0.731 | 0.054|
> > We do not find significant image quality difference between these two sets. Indeed after our CCS interpolation at the 0.07 rMSE target, the interpolated noise of not STG noise group all fall within 0.01 difference between zero mean and unit standard deviation. Empirically, at low rMSE levels, the noise may be non-standard Gaussian, but the samples are close to input image. At higher rMSE levels, the interpolated noise are more standard Gaussian resulting in good image quality.
>
> > To verify that more standard Gaussian distribution with more interpolation strength. We compute the average deviation of mean to 0, and deviation of variance to 1, for the CelebA-HQ experiment. We find as the interpolation strength $C_0$ increase, the deviation quickly narrows:
> | $C_0$ | deviation in mean | deviation in std |
> | ------| -----| -----|
> | 0.0 | 0.025 | 0.023 |
> |0.2  | 0.013 | 0.016 |
> |0.3  | 0.011 | 0.009 |
> |0.4  | 0.010 | 0.008 |
> |0.5  | 0.009 | 0.006 |
> |0.5  | 0.007 | 0.005 |
>
> > In our paper, our $C_0$ is mostly between 0.3 to 0.6, so we do not worry about the interpolated noise not standard Gaussian a lot. Also, we conduct experiments on different rMSE target level (with different $C_0$), and observe that increasing interpolation strength may lead to slightly better image quality if input image is not that good (the detailed experimental results can be found in R4's response). We will include the additional results in our revision.
>
> ***3. Q: More discussions on the inversion of DDIM not on Gaussian sphere***
> > 1. We will add a new proposition in our revision about more interpolation leads the second moment closer to a standard Gaussian. Formally: Fix any vector $\mathbf{x} \in \mathbb{R}^d$, and let $\boldsymbol{\epsilon} \sim \mathcal{N}(0, I_d)$. Let $\theta \in [0, \pi]$ be the angle between $\mathbf{x}$ and $\boldsymbol{\epsilon}$. We define the interpolated vector:
>
> > $\mathbf{y} = \frac{\sin(c\theta)}{\sin(\theta)} \mathbf{\epsilon} + \frac{\sin((1-c)\theta)}{\sin(\theta)} \mathbf{x}.$
>
> > Our goal is to show $\mathbf{y}$ is closer than $\mathbf{x}$ to a Gaussian in the second-moment (energy-shell) sense. Since a standard Gaussian has second moment $\mathbb{E}[\lVert Z \rVert^2] = d$, we define the gap of second moment as:
>
>
> > $\delta(Y) := \lvert \mathbb{E}[Y^2] - d \rvert.$
>
> > For any $c \in (0,1)$, we have:
>
> > $\delta(\mathbf{y}) \leq \delta(\mathbf{x}).$
>
> > Proof sketch: expand all the terms and apply trigonometric identities.
> In addition, some discussion about interpolating with Gaussian nosie or adding Gaussian noise to improve image sampling quality can be found in the literature [1,2].
>
> > 2. Furthermore, in the extreme case that the inverted noise is very not-Gaussian (for example a very blurry or noisy image input). We can perform Algorithm 5 in our paper. We observe that the sample image quality constantly improving by iteratively applying CCS interpolation sampling many times. In this sense, our method has the potential of performing image restoration, which is demonstrated in Fig.5 in our paper. Nevertheless, we acknowledge that non-standard Gaussian initial noise is a significant limitation of our work and will be better handled in future work.
>
>
> ***4. Q: Extending it to fine-grained image editing***
> > Thanks reviewer for pointing this direction! We are able to extend our CCS method to fine-grained image editing using null-space projection on the perturbation direction. The idea is that first we find the editing direction and then using Algorithm 4 in our paper for precise fine-grained editing. To do this, we can borrow the algorithm from LOCO-edit [3] The pseudo-code is listed below:
>
> > Step 1. Obtain mask of area of interest (for example excluding the background)
>
> > Step 2. Compute DDIM inversion of the input image with source prompt
>
> > Step 3. Follow LOCO-Edit [3] Algorithm 4 to obtain an local edit direction (here we need to tune the direction length by image quality)
>
> > Step 4. Now we have two initial noises, one by adding the local edit direction, the other is obtained through LOCO-Edit. We can then apply Algorithm 4 in our paper for precise local editing with user-specified editing strength.
>
> We perform experiments on this, which shows promising preliminary results. However, this is under development and will be investigated more thoroughly in future work.
>
> ***5. Q: Limitations not found***
> > Sorry for the confusion. We place the limitation section in the Appendix C, stating that our CCS algorithm does not purely focus on improving sampling quality but improving controllability while preserving sample quality from a novel and orthogonal noise space perspective. So there may be cases that other algorithms achieve better sample quality but with worse controllability. We will revise our paper to put it in our Conclusion section. On the other hand, we will include more limitations in our paper including:
>
> > DDIM inversion not perfectly standard Gaussian.
>
> > Some sampled images may be having issues like overlaying of two patterns or so on..
>
>
> [1] NoiseDiffusion: Correcting Noise for Image Interpolation with Diffusion Models. ICLR 2024
>
> [2] DreamClean: Restoring Clean Image Using Deep Diffusion Prior. ICLR 2024
>
> [3] Exploring Low-Dimensional Subspaces in Diffusion Models for Controllable Image Editing. NeurIPS 2024.

---

> ### Author Response · Authors · 2025-08-04
> **Revision of our manuscript based on reviewer's suggestions**
>
> Dear Reviewer ivEh,
>
>
> Thanks for your valuable feedback! We have revised our manuscript based on your suggestions. We highlight some of the core changes here:
>
> Below are our revisions:
> 1. In Section 4 (Controllable Sampling) and Section 5. We discuss about the alternative linear interpolation both theoretically and through experimental validation. (the addition method is the baseline we denoted as GP-C).
>
> >  First we theoretical demonstrate the problem with linear interpolation: in Line 205: We add one additional sentence: "On the other hand, a simple linear interpolation such as $\mathbf{x}'_T = a \mathbf{x}_T + (1 - a)\Delta_x$ for $0 < a < 1$ also cannot produce high-quality images", because this will shrink the magnitude of the interpolated vector. Formally, let $||\Delta x||_2 = ||x||_2$, and $\mathbb{E}[\Delta x] = 0$, we have $\mathbb{E} [ || \alpha x + (1-\alpha) \Delta x ||_2^2] < || x ||_2^2$. The proof can be found in Appendix A.3. "
>
> > Then, in section 5, we add the linear interpolation in our benchmarks for comparison. (FFHQ with pixel-based diffusion, CIFAR-10 with pixel-based diffusion and CelebA-HQ with Stable Diffusion 1.5). The new numbers are attached in the rebuttal.
>
> 4. On the problem with DDIM inversion may not lie on the standard Gaussian hypersphere.
>
> > In Section 4, we add another Proposition on more spherical interpolation leads the second moment closer to a standard Gaussian as Q.3 in our rebuttal.
>
> > In Section 5.2, we discuss 1) whether the inversion is far apart from the Gaussian sphere by validating on the CelebA-HQ and FFHQ datsets.  2) are sample quality compromised due to this inversion issue?. We add the results in Q.2 of our rebuttal to argue that with sufficient sample diversity, there is no significant change in sample quality, and the interpolated noise becomes more standard Gaussian rapidly as the interpolation scale $C_0$ increases
>
> > In Section 5.2, for the application of improving sample quality. We discuss the case when inverted noise is far from Gaussian sphere (i.e. very blurry or noisy input), we can apply our Algorithm 5 to correct the noise to become closer and closer to the Gaussian sphere, hence improving sample quality. We provide more visual example of using our CCS algorithm for image restoration in the Appendix.
>
> 5. Limitations: we revise our limitations in Section 6, line 344 to: "The limitation of our work includes 1. controlling statistical properties other mean and MSE is left as future work. 2. There might be some artifact samples that exhibit overlapping patterns. 3. DDIM inversion may not be perfectly standard Gaussian, which may hurt sample quality. Nevertheless, we demonstrate that our algorithm can remedy this issue." We have also added more visualizations to demonstrate our limitations in the Appendix.
>
> There are also two typos in our rebuttal (due to the tight rebuttal timeline).
>
> 1. When discussing DDIM inverted noise not on sphere. We measure the deviation of inverted noise from a zero mean and unit variance: the column $C_0$ should be 0.0, 0.2, 0.3, 0.4, 0.5, 0.6 instead of 0.0, 0.2, 0.3, 0.4, 0.5, 0.5.
> 2. When discussing extending to fine-grained image editing. the step 4 of our proposed algorithm should be: Now we have two initial noises, one is the inverted noise, and the other one is adding the local edit direction to the inverted noise.

---

> > ### Comment · Reviewer_ivEh · 2025-08-04
> >
> > Thanks for the detailed response. I have carefully read the rebuttal. Most of my concerns have been addressed.  I will keep my positive score. And I hope the authors could include the analysis about interpolation strategy and gaussian assumption in the final version.

---

> ### Author Response · Authors · 2025-08-04
> **Thank you**
>
> Dear Reviewer ivEh,
>
> Thank you for the encouraging feedback! We sincerely appreciate your valuable suggestion, which prompted us to reflect more deeply and broadly on the scope of our current work. We will devote more time studying the interpolation problem and the Gaussian sphere assumption and probably exploring whether there exists other indicator for sample quality such as likelihood computation through integration of score or so on.
>
> Your suggestion about finding certain localized directions: "keeping certain directions fixed while perturbing others more strongly." is inspiring and we are trying to extend our work in this direction. We are aware of the inherent spectral property of the Jacobian of denoiser of diffusion models (representing several semantic directions) [3], but the spectral property of the initial noise space still needs more exploration.
>
>
> [3] Exploring Low-Dimensional Subspaces in Diffusion Models for Controllable Image Editing. NeurIPS 2024.

---

### Official Review · Reviewer_6ZCA · 2025-07-02

**Clarity:** 2
**Significance:** 3
**Originality:** 2
**Rating:** 4
**Confidence:** 3

**Summary:**

Despite the current success of the Diffusion model, the connection between the initial noise perturbation $x_T$ and the resulting generated data $\bar{x}_0$ remains underexplored, limiting our understanding of how to control the sampling process. In this paper, the authors reveal that the change in generated outputs scales linearly with the magnitude of the initial noise perturbation. Building on this insight, the Controllable and Constrained Sampling (CCS) method is proposed to enable sampling with desired statistical properties while maintaining high sample quality.

**Questions:**

1. What is the relationship between the linear approximation of the score function and linearity in the output of Proposition 1? Why is the linear approximation error in output low when the score function is smooth (Lines 147-148)? Furthermore, should it be $o(||\lambda w||_2^2)$ instead  $o(\lambda)$? The proof A.1 needs to be carefully revised to enhance understanding, e.g., why not use $\lambda$ in Eq. 5 instead $\delta$ from scratch, the Taylor expansion should be revised too. Is the matrix-value function $\gamma_0$ bounded or not?

2. There are missing references in the appendix, e.g., [18] in the proofs of Proposition 2.  There might be an error in derivative of $\sigma(t)$ in the proof of Proposition 2 in Lines 820-821.

3. While the propositions are proved with $a=1$ and $b=\lambda$, do they still hold with $a \neq 1$?

4. In section 4.2, why can this approach not produce high-quality images when $x’_T$ is pushed farther from the spherical surface?

5. The paper only reports 1 selected MSE target value for each dataset. What happens when changing it? In addition to this, as the authors self-implement other baselines (Lines 315-317), are the corresponding hyperparameters of these baselines tuned?

6. Other minor points:
   - Typos: Line 79: $T$ -> $t$; Line 122: $x_t$ -> $x_T$; Line 160: $\alpha_0$ -> $\sqrt{\alpha_0}$;
   - repeated citation [2], [3]
   - Eq 4 needs further detail about the formula of $\bar{x}_t$ and $\sigma(t)$
   - Inconsistent format of some equations, e.g. $x$ in Line 134 versus others
   - Error in Proposition 1  e.g. $x_0(x_T)$ -> $x_0(x_T, T)$, I suggest the author should denote the $x_0(x_T, T)$ by another notation like denoising operator $D :x_T$ -> $x_0$
    - Algorithms 1 and 2 should be put side-by-side for improving representation.

**Ethical Concerns:**

["NO or VERY MINOR ethics concerns only"]

**Final Justification:**

Before the rebuttal, my main concerns centered around ambiguous arguments (as noted in my questions 1 and 4), issues with the empirical results (question 5), and an incorrect proof (question 2), along with several minor typographical errors. During the rebuttal, the authors made a commendable effort to address these concerns and proposed concrete revisions to improve the quality of the paper. Therefore, I have decided to increase the score from 2 to 4, on the condition that the authors implement the suggested revisions in the final version.

**Limitations:**

Please see the above questions.

**Quality:**

2

**Strengths And Weaknesses:**

Strengths:
- The paper uncovers a novel linear relationship between the initial noise and the generated samples under DDIM sampling, offering new insights into the diffusion sampling process.

- It introduces a new task of controllable generation, which aims to match the sample mean to a target mean while constraining the sample MSE to a desired level. To address this task, the authors propose an original controllable sampling method.

Weaknesses

- The writing quality requires improvement to enhance clarity and readability throughout the paper.

- Several key aspects of the methodology and presentation are confusing, as elaborated in the Questions section below.

---

> ### Author Rebuttal · Authors · 2025-07-27
>
> We thank the reviewer for the careful and detailed review!
>
> Here we provide the answers to the questions:
> 1. ***Q: What is the relationship between the linear approximation of the score function and linearity in the output of Proposition 1***:
> > **A**: The way we prove Proposition $1$ is by recursively using the linear approximation of the score function. In more details,
> since each DDIM sampling step takes a linear combination of the predicted score and the intermediate noise, we prove that with the assumption of linear approximation of score function, we can derive the linearity behavior of DDIM sampling output and input. Formally as in the proof of Proposition 1 (in Appendix Line 805), let $L_T$ be a one step DDIM sampling step on x, and $\eta_T$, $\lambda_T$ be DDIM sampling constants, $w$ be a unit direction vector, and $\delta$ be the perturbation scale, we have $L_T(x + \delta w) = \eta_T (x + \delta w) + \lambda_T \nabla_{x}\log p_{T}(x + \delta w) = \eta_T x + \lambda_T \nabla_{x}\log p_{T}(x) + \lambda_T \delta H_T(x)w + \delta\eta_T w+ o(\delta)$. This demonstrates that for one-step DDIM sampling, the linearity between the output and the input is proportional to the linear approximation error of the score function. Then we can further extend this to the entire DDIM sampling process as demonstrated in line 808-813 in the Appendix. We will revise our paper to clarify this relationship and describe this reasoning process clearly.
>
> 2. ***Q: Why is the linear approximation error in output low when the score function is smooth?***
> > **A:** When a function $f(x)$ is smoother, it has lower magnitude for the higher-order derivatives (second-order or higher). Intuitively, it has fewer abrupt changes. So when zooming in on the plot of $f(x)$ versus $x$, a smoother function appears more like a straight line locally.
>
> >  In the Appendix B.3. (Line 896-919), we have provided a theoretical analysis of this linear approximation error, and derived the bound for this error. Based on reviewer’s comment to improve the clarity, we will revise our paper to include the Taylor remainder theorem in the main text, and provide a more thorough reasoning and derivation as below to support the previous claim: less smoothness comes with larger magnitude of higher-order (>2) gradients, which introduces more local nonlinearity.
>
> > Formally, we can use the Taylor remainder theorem here:
> >
> > $$R_n(x) = \frac{f^{(n+1)}(c)}{(n+1)!} (x - a)^{n+1}$$
> > This theorem demonstrates that the linear approximation error is bounded by the magnitude of its second-order derivative at the point of approximation, which explains why a smoother function has less linear approximation error.
> >
> > Take three functions as examples:
> >
> > 1. $f(x) = |x|$: This function is not differentiable and not smooth at $x = 0$ because it has a sharp corner. Using sub-gradient for linear approximation leads to large error.
> >
> > 2. $f(x) = e^x$: This function is smooth everywhere, with all derivatives equal to $e^x$. The linear approximation error at $x = 0$ is very small and the linear approximated value is very close to the true function value because the higher-order derivatives are well-behaved and continuous.
> >
> > 3. $f(x) = \sin(\frac{1}{x^2 + 0.0001})$: Even though this function is differentiable, it has rapid oscillation at x = 0 (much less smoother than $e^x$), Its linear approximation at $x = 0$ is very inaccurate and very different from the true value of the function.
>
> 3. ***Q: Is $\gamma_0$ bounded?***:
> >If we assume the Hessian of $\log p_t(x_t)$ is bounded (it changes less abruptly), then it is bounded.
>
> 4. ***Q: Should it be $o(||\lambda w||^2_2)$ instead of $o(\lambda)$? why not use $\lambda$ in Eq. 5 instead $\delta$ from scratch?***
> > There seems to be a confusion on the notation. First, from line 805 to line 808, we refer to an arbitrary fixed-direction perturbation with a unit length $w$, so we use the notation $\delta$ as the scale of that direction. In Eq.5, we refer to a specific unit-length perturbation to the initial noise in DDIM sampling, so we use a different syntax. We realize that this notation system can be confusing, and we will add more context in our revision to explain these notations clearly in the Appendix.
>
> > To our best knowledge, the error term in Proposition 1 should be $o(\lambda)$. The reason we exclude $w$ in the error term is that we have $||o(\lambda) w || \leq o(\lambda) ||w || \leq o(\lambda)$ since we denote $w$ as a unit direction vector as in line 805. In line 810, we get that $x_0(x_T + \lambda \Delta x, T) = x_0(x_T) + \lambda \gamma_0(x_T) \Delta x + o(\lambda)$, by rearranging the terms, we now have $x_0(x_T + \lambda \Delta x, T) - x_0(x_T) = \lambda \gamma_0(x_T) \Delta x + o(\lambda)$. Now let's apply the triangle inequality of the norm twice. First:
> > 1. we notice that $||x_0(x_T + \lambda \Delta x, T) - x_0(x_T)  ||_2= ||\lambda \gamma_0(x_T) \Delta x + o(\lambda)||_2 \leq ||\lambda \gamma_0(x_T) \Delta x|| + ||o(\lambda)||_2$
> > 2. we also notice that  $||\lambda \gamma_0(x_T) \Delta x||_2 \leq ||x_0(x_T + \lambda \Delta x, T) - x_0(x_T) ||_2 + ||o(\lambda)||_2$
> >Combine these two inequalities, we have $||\lambda \gamma_0(x_T) \Delta x||_2 - ||o(\lambda)||_2 \leq ||x_0(x_T + \lambda \Delta x, T) - x_0(x_T) ||_2 \leq ||\lambda \gamma_0(x_T) \Delta x||_2 + ||o(\lambda)||_2$, so we can conclude that $||x_0(x_T + \lambda \Delta x, T) - x_0(x_T) ||_2 = \lambda|| \gamma_0(x_T) \Delta x||_2 + o(\lambda)$.
>
> > Indeed, $o(||\lambda w||^2_2)$ is tighter than our current error bound. We do not achieve this tight bound in our paper, and will investigate this in future work. We will clarify the notation in our revision.
>
> 5. ***Q: There might be an error in derivative of $\sigma_t$***
> > Thanks!  There is a typo in line 820. There should be an additional $\frac{1}{\alpha(t)^2}$ term in the derivative of $\sigma(t)$. We will revise the $h(t,y)$ in line 815, and line 820. Nevertheless, adding this term does not affect the assumption of continuous derivative since $\alpha(t)$ is smooth function that is larger than zero between [0,T], so that other places in the proof and the conclusion is not affected. We will revise the proof in our revision.
>
> 6. ***Q: Does the Propositions still hold for $a \neq 1$?***
> > Only Proposition 5 has the setting of $a = 1$, and $b = \lambda$, while Proposition 6 has a different spherical setting. Proposition 5 just explains why using $a = 1$, and $b = \lambda$ not working. We can also demonstrate using simple linear interpolation ($a \neq 1$) not working for sampling as well:
> Formally, let $||\Delta x||_2 = ||x||_2$, and its direction is uniformly distributed, and $\mathbb{E}[\Delta x] = 0$, for $0 < \alpha < 1$, we have
> > $$\mathbb{E} [ || \alpha x + (1-\alpha) \Delta x ||_2^2] < || x ||_2^2$$
>
> > Proof: we have  $||\alpha x + (1-\alpha) \Delta x ||_2^2 = \alpha^2 || x ||_2^2 + 2\alpha(1-\alpha)(x \cdot \Delta x) + (1-\alpha)^2 || \Delta x ||_2^2$, since we have $\mathbb{E}[\alpha(1-\alpha)(x \cdot \Delta x)] = \alpha(1-\alpha) x \cdot \mathbb{E}[\Delta x] = 0$, we have $\mathbb{E} [ || \alpha x + (1-\alpha) \Delta x ||_2^2] = \alpha^2 || x ||_2^2 +  (1-\alpha)^2 || \Delta x ||_2^2 = (\alpha^2 + (1-\alpha)^2) || x ||_2^2$, now since $2\alpha (\alpha - 1) < 0$, we have $(\alpha^2 + (1-\alpha)^2) < 1$, and we can reach the conclusion that using simple linear interpolation cannot preserve the norm leading to falling apart from the Guassian sphere.
>
> > We will include this analysis as well as experimental results on simple linear interpolation (also mentioned by R3) in our revision.
>
> 7. ***Q: Why can this approach not produce high-quality images when is pushed farther from the spherical surface?***
> > Intuitively, if initial noises not on spherical surface, this means that the noise level is either too low or too high by Proposition 4 in our paper, which both leads the trained denoiser of the diffusion model not effectively removing the noise effectively (noisy image) or removing too much noise (blurry image). Only if we ensure the initial noise level matches the pre-defined noise level at T, the diffusion model can sample as it is designed to. This phenomenon is also observed in [1], and a qualitative example is demonstrated in Fig.3 in that paper.
>
> > We will revise our paper to include an explanation and qualitative examples on why sampling quality degrades with initial noise not on the spherical surface.
>
> 8. ***Q: What happens with different MSE targets? How are baselines tuned?***
> > Thanks for the suggestion! The target in this paper is set for best visualization quality (both preserving source image feature and having variations). We perform additional experiments with sampling quality benchmark with rMSE targets from [0.05, 0.06, 0.07,0.08,0.09, 0.10] for Stable Diffusion on CelebA-HQ. We observe that our method consistently performs quite well (PSNR is for sample mean v.s. target image, SD is for diversity). (more numbers in R4's response).
>
> > We tuned hyperparameters based on validation set. More details can be found in Appendix D.1 and R1 response. Indeed most hyperparemeters of our method and baselines are searched by our controller. We will revise our paper to include all baseline details.
>
> (numbers are reported in the order of CCS-C/CCDF-C (the strongest baseline))
> | Target rMSE | PSNR↑  | MUSIQ↑ | CLIP-IQA↑| SD↑ |
> | ---------| ---- | ----- | -------| -----|
> |0.05 | 32.22/30.86   | 49.60/49.53 | 0.729/0.730 | 0.036/0.034|
> | 0.06 | 31.44/29.03| 49.58/49.02 | 0.731/0.731| 0.043/0.040|
> | 0.07 | 30.29/27.66 | 49.66/48.91 | 0.732/0.735| 0.053/0.051|
> | 0.08 | 30.10/26.80| 49.85/48.23 | 0.742/0.729| 0.056/0.052|
> | 0.09 | 29.74/25.98 | 49.82/48.01 | 0.740/0.732| 0.061/0.054|
> | 0.10 | 29.31/25.20 | 49.80/46.74 | 0.731/0.727| 0.063/0.057|
>
> 9. ***Q: Minor issues***
> > Thanks! We will revise all points.
>
> [1] NoiseDiffusion: Correcting Noise for Image Interpolation with Diffusion Models. ICLR 2024

---

> ### Author Response · Authors · 2025-08-04
> **Revision of our manuscript based on reviewer's suggestion**
>
> Dear Reviewer 6ZCA,
>
> Thanks for your valuable suggestions. Highlights of our revisions:
>
> 1. Below the statement of Proposition 1. We add one line at Line 141:
> "Recalling Eq.2, each idealized DDIM sampling can be viewed as a linear combination of the current intermediate noisy input $x_t$ and the score function $\nabla_{x_t} \log p_{t}(\mathbf{x})$. Based on this observation, proposition 1 can be derived through recursively using the linear approximation of the score function since each DDIM sampling step takes a linear combination of the predicted score and the intermediate noise. The derivation can be found in Appendix A.1."
>
> 2. At Line 147, we modify the sentence ```Furthermore, when at large $t$, $p_t$ is approximately Gaussian and $\nabla x_t \log p_t(x_t)$ is smooth, which leads to low linear approximation error``` to
> "Furthermore, at $t$, $p_t$ is approximately Gaussian and $\nabla x_t \log p_t(x_t)$ is smooth, which implies small linear approximation error. The reason for the small error is that 1) the score function of Gaussian is linear in $x$ by $\nabla_x \log p(x) = -\Sigma^{-1}(x - \mu)$, 2) When a function is smooth and its higher-order derivatives are small in magnitude, it has fewer abrupt changes and the linear approximation error is bounded by the norm of the Hessian of the score function (through Taylor Remainder Theorem), leading to low linear approximation error."
>
> 3. Proof of Proposition 1 (to save space, we sometimes use $x$ , but it should be $\mathbf{x}$.)
>
> > 3.1: Line 805 in Appendix: we revise the sentence: "for any fixed direction $w$" to "for any fixed direction  of unit length $w$".
>
> > 3.2: For the matrix-value function in Line 807. We revise $\gamma_T(\mathbf{x})$ is a ```matrix-valued function``` to "matrix-value function which is bounded if the norm of the Hessian of $\log p_t$ is bounded (which is usually the case when the log-likelihood does not change abruptly). "
>
>
> > 3.3: Line 810, we add more lines for transition from $\delta$ to $\lambda$:
> >"By applying the recursion T times, we have
> $$\mathbf{x}_0(\mathbf{x} + \delta \mathbf{w}, T) := L_0 \circ \cdots \circ L_T(\mathbf{x}) + \delta \gamma_0(\mathbf{x})\mathbf{w} + o(\delta)$$
> >Now let $\lambda$ be the scale of the perturbation, such that $\lambda > 0$ and $ \lambda \in \mathbb{R}$, and let $\Delta x$ be the unit-length perturbation to the initial noise $\mathbf{x}_T$, we have"
> $$
> \mathbf{x}_0 (\mathbf{x}_T + \lambda \Delta \mathbf{x}, T) = \mathbf{x}_0 (\mathbf{x}_T, T) + \lambda \gamma_0(\mathbf{x}_T) \Delta\mathbf{x} + o(\lambda)
> $$
>
> > 3.4: At the end of the proof (line 811), we add "after obtaining Eq.5, we apply the triangle inequality of norm twice, which gives us"
> $$
> || \lambda \gamma_0(x_T) \Delta x ||_2 - || o(\lambda) ||_2
> \leq || x_0(x_T + \lambda \Delta x, T) - x_0(x_T) ||_2
> \leq || \lambda \gamma_0(x_T) \Delta x ||_2 + || o(\lambda) ||_2
> $$
>
>
> 4. Proof Proposition 2
>
> > 4.1: We added the citation [2] ([18] in our paper) to the Appendix A.2.
>
> > 4.2: Thanks for catching the typo. We revise Line 815 to be
> $h(t, \mathbf{y}) := -\frac{1}{2} \frac{\sqrt{\alpha(t)}}{\alpha(t)^2}\alpha'(t) \nabla \log p_t (\frac{\mathbf{y}}{\sqrt{\sigma^2(t) + 1}})$, and then proof of Proposition 2 be $$ \text{d} \bar{\mathbf{x}}_t = \frac{1}{2} \frac{\sqrt{\alpha(t)}}{\alpha(t)^2}\alpha'(t) \nabla \log p_t (\frac{\mathbf{y}}{\sqrt{\sigma^2(t) + 1}}) = h(t, \mathbf{y}) \text{d} t$$.
>
> 5. Explanation of necessity of initial noise falling on spherical surface.
> Line 188: We add one additional sentence:
> "Intuitively, if initial noises are not on spherical surface of standard Gaussian, then the noise level can be either too low or too high, which both lead the trained denoiser of the diffusion model not effectively removing the noise effectively or removing too much noise. A similar observation is also reported in [1]."
>
> 6. Adding discussion on $a \neq 1$ (e.g. linear interpolation).
> Line 205: We add one additional sentence: "On the other hand, a simple linear interpolation such as $\mathbf{x}'_T = a \mathbf{x}_T + (1 - a)\Delta_x$ for $0 < a < 1$ also cannot produce high-quality images", because this will shrink the magnitude of the interpolated vector. Formally, let $||\Delta x||_2 = ||x||_2$, and $\mathbb{E}[\Delta x] = 0$, we have $\mathbb{E} [ || \alpha x + (1-\alpha) \Delta x ||_2^2] < || x ||_2^2$. The proof can be found in Appendix A.3. "
>
> 7. Line 79: we revised "The conditional score at T" to "The conditional score at t".
>
> 8. Line 160 $\alpha (0)$ to $\sqrt{\alpha(0)}$
>
> 9. Line 116: we add "Let $\bar{\mathbf{x}}_t$ be a continous variable representing $x$ across $t$ and let $\sigma(t) = \sqrt{(1- \alpha(t))/\alpha(t)}$".
>
> 10. Line 134 and all other places, we bold $x$.
>
> 11. Literature in diffusion models sometimes uses $\mathbf{x}_0(\mathbf{x}_T)$, we change all $\mathbf{x}_0(\mathbf{x}_T)$ to $\mathbf{x}_0(\mathbf{x}_T, T)$.
>
> [1] NoiseDiffusion. ICLR 2024.
>
> [2]. Hartman, P. Ordinary differential equations. SIAM, 2002.

---

> > ### Comment · Reviewer_6ZCA · 2025-08-06
> > **Rebuttal response**
> >
> > Thank you for the thorough and thoughtful responses to my concerns. Your explanations have addressed most of the issues I raised, and I appreciate the effort you put into clarifying them. As a result, I am increasing the score to 4.
> >
> > However, I strongly encourage the authors to revise several aspects of the paper, as mentioned above, to improve clarity and ensure that the contributions and methodology are presented in a more accessible and reader-friendly manner.

---

> ### Author Response · Authors · 2025-08-06
>
> Dear Reviewer 6ZCA,
>
> Thank you for the encouraging feedback! We really appreciate your careful and detailed inspection of our manuscript. We agree that we should be more careful and prudent when it comes to writing: to make each paragraph clear and self-containing. We will try our best to revise our manuscript for correctness and best readability. We are thankful for the reviewer catching some typos such as the proof of Proposition 2 and notations on $\mathbf{x}$.
>
> In addition, we thank the reviewer for finding some cases we have not discussed such as changing target MSE levels, when $a \neq 1$, and so on. These cases are essential for improving our paper.

---

### Official Review · Reviewer_ZiPM · 2025-07-03

**Clarity:** 3
**Significance:** 2
**Originality:** 3
**Rating:** 4
**Confidence:** 4

**Summary:**

This paper proposes CCS, a training-free method to guide diffusion model outputs toward target statistical properties by operating on the initial noise. The authors identify a linear relationship between initial noise perturbations and final outputs under DDIM sampling. The paper offers theoretical analysis, empirical validation, and applications in controllable sampling and precise image editing.

**Questions:**

1. How does the linearity relationship hold in complex or highly multimodal distributions?
2. Can CCS be combined with per-sample controls like classifier guidance or universal guidance?
3. How sensitive is CCS to the choice of timestep in full vs. partial inversion?

**Ethical Concerns:**

["NO or VERY MINOR ethics concerns only"]

**Final Justification:**

I thank the authors for the detailed and thoughtful rebuttal and for revising the manuscript. I appreciate the additional experiments on multimodal and out-of-distribution data, as well as the clarifications on combining CCS with classifier guidance and the effects of inversion timestep. I also noticed other concerns raised by fellow reviewers have also been nicely addressed. Thus I lean towards acceptance for this work.

**Quality:**

3

**Strengths And Weaknesses:**

Strength:

1. The linearity identified in DDIM sampling between the initial noise and the output is interesting.
2. The authors have provided a rigorous analysis using ODE theory and validated it with extensive experiments.
3. I appreciate the writing of this work with strong algorithmic and experimental clarity.

Weakness:

1. The generalization of linearity across diverse distributions (e.g., highly multimodal datasets) is not thoroughly examined. I wonder this part.

---

> ### Author Rebuttal · Authors · 2025-07-30
>
> We thank the reviewer for the careful and insightful review!
>
> ***1. Q: How does the linear relationship hold in complex or highly multimodal distributions?***
>
> > We thank the reviewer for this inquiry, which is also a question that we are very interested in. We include more results in this rebuttal and will include these new results in our revision. By our analysis of the linear approximation error (in Appendix B.3), we bound the error by the reciprocal of the lowest probability density in a local region, and we also bound the linear approximation error by the (higher-order) gradients of the probability distribution of $p_t(x_t)$. We conjecture that for multimodal distributions the linearity will be weaker at the low-density regions (regions between each modalities).
>
> > To further investigate this relationship, we perform additional experiments on three more datasets to test how the linearity change for multimodal distributions. We are interested in
> > 1. How is the linearity when the model is trained on multimodal data and sample with out-of-distribution target image from another highly multimodal dataset? or an OOD simple dataset?
> > 2. How is the linearity when the model is trained on a simple dataset (for example face dataset like FFHQ), and sample on out-of-distribution multimodal dataset? on OOD simple dataset?
> >3. How is the linearity when training on simple dataset and sample on the same dataset?
> >4. How is the linearity comparing model trained on multimodal dataset testing in-distribution and model trained on simple dataset testing in-distribution?
>
> >In our experiment section of our paper, we cover partially 1,3, and 4: we test pixel diffusion models on FFHQ dataset, and the CIFAR dataset. We also test the Stable Diffusion model (1.5) [1]  which is trained on a complex dataset LAION-5B, on human face dataset (CelebA-HQ). We observe that diffusion models trained on simple dataset (FFHQ) and testing in-distribution exhibits significant stronger linearity than the Stable Diffusion model on the CelebA-HQ dataset.
>
> >We observed in our paper that linearity decreases slightly when comparing diffusion model with multimodal training data to diffusion model with simple training data. CIFAR-10 is a dataset with different classes, we have slightly lower linearity score for CIFAR-10 compared to FFHQ even though CIFAR-10 is of lower resolution. Empirically, we observe sudden change of image semantics occasionally.
>
> > In this rebuttal, we provide additional results for 1,2, and 4. We include three more datasets:
> > 1. We pick 4 images from the class 0, 4, ...,99 from the validation set of ImageNet [2]
> > 2. We pick five videos from the classes: "Applying Eye Makeup", "Baby Crawling", "Billiard", and "Blow Dry Hair", and sample five frames from each video in the UCF-101 dataset [3]
> > 3. We pick 10 images from each organ site in the AAPM dataset as in [4], which is CT scans of different body parts.
> These datasets are different from the training data of SD1.5. We use the same linearity testing methods as in Section 5.1 in our main paper (also in Appendix B. We first summarize the results below for Stable Diffusion 1.5:
> |Dataset | ImageNet | UCF-101| ```CelebA-HQ``` | ```fMoW``` |
> |-----------|------| --------| --------| ------------|
> | R-square of output change v.s. perturbation scale | 0.960 | 0.962 | 0.959 | 0.947|
> | cosine similarity of linear combinations | 0.922 | 0.924 | 0.901 | 0.920 |
>
> > The results show that for multimodal dataset (containing many classes like ImageNet or UCF-101), there is no evidence of decrease in linearity compared to simple datasets (CelebA-HQ) for the same backbone model trained on a large multimodal dataset (LAION-5B). The less linearity on CelebA-HQ may be due to our data processing techniques (upscale 256 to 512 introduces bluriness and let initial noise apart from the Gaussian sphere).
>
> > When testing the linearity on OOD dataset for model trained on simple dataset, we observe significant linearity drop.  We  compare it model trained on multimodal data and tested on simple dataset. Results below use the same testing method and dataset as mentioned before and in the paper Appendix B. Note that the trained->tested here mean that the model is trained on the training set of "trained" and tested on the validation set of "tested".
> |Dataset (trained->tested) | ImageNet->FFHQ | FFHQ->ImageNet | FFHQ->FFHQ | ImageNet->ImageNet | FFHQ->AAPM|
> |-----------|------| --------| --------| ------------|-----------|
> | R-square of output change v.s. perturbation scale | 0.934 | 0.938 | 0.995 | 0.983 | 0.951 |
> | cosine similarity of linear combinations | 0.902 | 0.905 | 0.958 | 0.941 | 0.912 |
>
> > For fair evaluation we use the same model architecture DDPM++ with the same training loss for these two pixel-space diffusion models. We find that for not foundation models, the OOD linearity drops significantly. The multimodal backbone has a slightly lower linearity score than the simple data backbone due to the complexity of its training data (it learns abrupt changes). The drop in OOD linearity is for both multimodal data and simple data, suggesting that the probability density of input image plays an important role in linearity, in agreement with our theoretical analysis in Appendix B.3.
>
> > Overall, we observe that this linearity exists for different settings, and changes based on the probability density of input data and the curvature/smoothness of the training data distribution. OOD data tends to have lower linearity score. backbones trained on multimodal data may have lower linearity score. In addition, well-trained diffusion models trained on the same dataset should have similar linearity score by the reproducibility of diffusion model in [5] (they learn the same probability density which determines linearity).
>
>
>
> ***2. Q: Can CCS be combined with per-sample controls like classifier guidance or universal guidance?***
>
> > Yes, it can be combined with per-sample control method such as CFG (other guidance method should also be fine). We just need to replace the score with the CFG score to perform CCS sampling. Indeed combining with CFG yields very interesting results. We will add new results to our revision.
> We perform additional experiments testing the constrained sampling with the target rMSE of 0.07 on the CelebA-HQ dataset with the SD1.5 checkpoint with different CFG levels [1.5, 2.5, 3.5, 4.5] (we perform inversion using this specific CFG value and then perform sampling with the same CFG value.) The starting t is set to be 45 which is consistent with our main experiments. The results are summarized in the table below (MUSIQ is for image visual quality, CLIP-IQA for semantic quality, SD for diversity and PSNR for controllability):
>  | CFG | PSNR↑  | MUSIQ↑ | CLIP-IQA↑ | SD↑ |
> | ---------| ---- | ----- | -------| -----|
> |1.5 | ***30.19***  | 49.70| 0.733 | 0.056 |
> | 2.5 | 29.98 | 51.28 | ***0.737*** | ***0.057***|
> | 3.5 | 29.15 | 51.29| 0.730| 0.052|
> | 4.5 | 27.28 | ***52.85*** | 0.729| 0.053|
> We observe that
> >1. When CFG gets higher, the controllability decreases, but images get sharper with better visual quality
> >2. There seems to be a sweet spot for the diversity. While high CFG hurts sampling diversity, medium-level may be not bad.
>
> We also want to note that other baselines can fail at high CFG. CCDF-C (Our strongest baseline) uses SDEdit-type noise adding, so it cannot preserve source image information at a large rMSE target (it always generate CFG-guided image). At CFG=4.5, we have PSNR at 25.73 and SD of 0.079 at rMSE target 0.1 for our method, but CCDF-C drops to 20.19 and SD of 0.032, which signals significant loss of controllability and diversity.
>
>
> ***3. Q: How sensitive is CCS to the choice of timestep in full v.s. partial inversion?***
>
> We perform additional experiments to test the different inversion timesteps on the Stable Diffusion 1.5 for timestep [37, 40, 42, 45, 48] on the CelebA-HQ dataset. Results show that small timestep leads to better controllability, but worse diversity and slightly worse image quality.
>
>  | Inversion Timestep | PSNR↑  | MUSIQ↑ | CLIP-IQA↑ | SD↑ |
> | ---------| ---- | ----- | -------| -----|
> |37 | 31.23 | 49.19 | 0.729 | 0.041 |
> | 40 | ***31.28*** | 49.31 | 0.732 | 0.047|
> | 42 | 30.78 | 49.53 | 0.730| 0.049 |
> | 45 | 30.29 | 49.66 | ***0.732*** | ***0.053***|
> | 48 | 30.15 | ***49.87*** | 0.731| 0.052|
>
> We conjecture that editing image is harder with smaller inversion timesteps, so there is a diversity drop but with a better controllability. Also, more denoising steps can improve the sample quality. We will add these new results to our revision.
>
>
> [1] High-resolution image synthesis with latent diffusion models. CVPR 2023
>
> [2] Imagenet: A large-scale hierarchical image database. CVPR 2009
>
> [3] Ucf101: A dataset of 101 human actions classes from videos in the wild.
>
> [4] Piner: Prior-informed implicit neural representation learning for test-time adaptation in sparse-view ct reconstruction. WACV 2023
>
> [5] The emergence of reproducibility and consistency in diffusion models. ICML 2025

---

> ### Author Response · Authors · 2025-08-04
> **Revision of our manuscript based on reviewer's suggestion**
>
> Dear Reviewer ZiPM,
>
> Thanks for your valuable suggestions. We revise our paper based on your suggestion. We highlight some core changes here:
>
> 1. In Section 5.1, we include a discussion about the linearity on multimodal distributions which includes new results in the rebuttal. "To further investigate how the linearity changes with the complexity of dataset, and different diffusion model backbones. We perform experiments to test1. pretrained diffusion models on a simple dataset such as FFHQ. 2. pretrained diffusion models on multimodal dataset (with many classes) such as ImageNet [1]. 3. large pretrained foundation models such as Stable Diffusion Model (trained on complex multimodal dataset). We hypothesize that the linearity on out-of-distribution dataset will decreases, so we test the pretrained pixel-diffusion models on OOD datasets. While the training data for SD1.5 is very large (LAION-5B), we just test it on other multimodal datasets such as UCF-101 [2], and Imagenet [2]. Results show that 1. The linearity decrease significantly when testing on OOD dataset for pixel-diffusion models. 2. For foundation models, complexity of dataset for sampling does not affect the linearity significantly. 3. The linearity decreases slightly comparing diffusion models trained on multimodal dataset to those trained on simple dataset."
>
> 2. We provide discussion on CFG and adjusting the choice of timestep in the Appendix. We observe that 1. When CFG gets higher, the controllability decreases, but images get sharper with better visual quality. 2. High CFG levels hurt diversity but medium level is fine. 3. Smaller timesteps lead to better controllability, but worse diversity and slightly worse image quality.

---

> > ### Comment · Reviewer_ZiPM · 2025-08-05
> >
> > I thank the authors for the detailed and thoughtful rebuttal and for revising the manuscript. I appreciate the additional experiments on multimodal and out-of-distribution data, as well as the clarifications on combining CCS with classifier guidance and the effects of inversion timestep. These additions strengthen the empirical results. I will keep my current score.

---

> ### Author Response · Authors · 2025-08-06
>
> Dear Reviewer ZiPM,
>
> Thank you for the encouraging feedback! We are grateful for you raising important questions such as how linearity holds in multimodal dataset and how the performance of our method with CFG guidance is. We are actively investigating the linearity property and associating it with diffusion model distillation, post-training efficiency and many other directions. In addition, we will perform more thorough study on CFG guidance and extend our theoretical analysis with CFG guidance in future work.

---

### Note · Authors · 2025-08-12

We sincerely thank all reviewers for their careful reading and constructive feedback. All reviewers have indicated that our clarification and additional experiments addressed their main concerns, without unresolved issues. We believe our work has sufficient technical contribution and novelty, as supported by the reviewers’ positive feedback.

***Summary of Rebuttal***

1. ***Linearity analysis across diverse distributions***:
We extended our evaluation to multimodal datasets (ImageNet, UCF-101) and out-of-distribution (OOD) cases. Results confirm that the linearity property persists across diverse datasets or different backbones, while there exists a linearity drop for OOD datasets and for backbones trained on multimodal data, consistent with our theoretical analysis.

2. ***Significance of the controllable sampling problem setting***:
We clarified that our goal is to sample high-quality images at a specified target diversity level while preserving the core features of the input image. This setting enables fair comparisons of personalized image generation and editing quality. To the best of our knowledge, few prior works explore this direction. Our two key ideas are: (1) controlling the average distance (rMSE) between samples and the input image, and (2) measuring the distance between the sample mean and the target image, which enable us to achieve this goal.

3. ***Efficiency & downstream applicability***:
We clarified that our CCS sampling is among the most time-efficient methods comparing to baselines. We highlighted downstream uses, including precise image editing, personalized album generation and image restoration from degraded inputs.

4. ***Results on different rMSE targets***:
We conducted additional experiments across a wide range of rMSE target levels, showing that our method consistently outperforms baselines in controllability (~10% better) and diversity while maintaining equal or superior image quality. Furthermore, increasing the rMSE target seems not to cause a decline in sample quality.

5. ***DDIM-inverted noise may not lie on a Gaussian hypersphere***:
We demonstrated, both empirically and theoretically, that spherical interpolation can bring DDIM-inverted noise distributions closer to standard Gaussian, without degrading image quality when the initial noise is non-standard Gaussian.

6. ***Revision of the manuscript***:
We revise our manuscript for enhanced clarity with new results. We highlighted major changes in our response below.

---

### Decision · Program_Chairs · 2025-09-17

**Decision:**

Accept (poster)

**Comment:**

* Summary
Authors propose a new method to guide diffusion models in a training-free way, denoted as CCS. The method relies on performing a conformal/near-linear transformation on the noise factor itself, rather than performing a diffusion process on the data explicitly. This actually sheds light on not just the noise but how the entire noise distribution gets changed under the dynamics through linear steps.

* Strengths and reasons for acceptance
The paper addresses an essential aspect of diffusion models, the actual noise dynamics, which is novel in both its examination and by providing tractable analytical arguments as well as empirical evidence on several datasets. It's also, to the best of my knowledge, the first time someone does that by moment matching on the noise function. On the empirical side, the authors (after rebuttal) also performed a thorough ablation test of hyperparameters. The paper (in the appendix) also gives all of the required theoretical proofs for the results and statements made in the main paper. All of the above makes me recommend this paper for a weak accept. I have no issue in bumping this down if needed, especially given the weakness.

* Weaknesses
The paper is not as clear or well-written, while the authors have raised the level post rebuttal, which is still contingent on integration post rebuttal comments. While authors do give some of the experimental results, they are still on a pretty small dataset, and more importantly, the examples that are shown visually seem a bit cherry-picked, and there is no negative example shown.